# Gendered hiring and attrition on the path to parity for academic faculty

Nicholas LaBerge[1]*, Kenneth Hunter Wapman[1], Aaron Clauset[1,2,3], Daniel B Larremore[1,2,3]*

[1]Department of Computer Science, University of Colorado, Boulder, United States; [2]BioFrontiers Institute, University of Colorado, Boulder, United States; [3]Santa Fe Institute, Santa Fe, United States

**Abstract** Despite long-running efforts to increase gender diversity among tenured and tenure-track faculty in the U.S., women remain underrepresented in most academic fields, sometimes dramatically so. Here, we quantify the relative importance of faculty hiring and faculty attrition for both past and future faculty gender diversity using comprehensive data on the training and employment of 268,769 tenured and tenure-track faculty rostered at 12,112 U.S. PhD-granting departments, spanning 111 academic fields between 2011 and 2020. Over this time, we find that hiring had a far greater impact on women's representation among faculty than attrition in the majority (90.1%) of academic fields, even as academia loses a higher share of women faculty relative to men at every career stage. Finally, we model the impact of five specific policy interventions on women's representation, and project that eliminating attrition differences between women and men only leads to a marginal increase in women's overall representation—in most fields, successful interventions will need to make substantial and sustained changes to hiring in order to reach gender parity.

**\*For correspondence:**
nila1617@colorado.edu (NLaB); daniel.larremore@colorado.edu (DBL)

**Competing interest:** The authors declare that no competing interests exist.

## eLife assessment

Efforts to increase the representation of women in academia have focussed on efforts to recruit more women and to reduce the attrition of women. This study - which is based on analyses of data on more than 250,000 tenured and tenure-track faculty from the period 2011-2020, and the predictions of counterfactual models - shows that hiring more women has a bigger impact than reducing attrition. The study is an **important** contribution to work on gender representation in academia, and the evidence in support of the findings is **convincing**.

## Introduction

Faculty play a crucial role in educating future researchers, advancing knowledge, and shaping the direction of their fields. Diverse representation of social identities and backgrounds within the professoriate improves educational experiences for students (*Lockwood, 2006*; *Stout et al., 2011*), accelerates innovation and problem-solving (*Hofstra et al., 2020*; *Page, 2008*; *Yang et al., 2022*), and expands the benefits of scientific advances to a broader range of society (*Koning et al., 2021*; *Kozlowski et al., 2022*). Gender equality in particular is a foundational principle for a fair and just society in which individuals of any gender identity are free to achieve their full potential. However, the composition of the professoriate has never been representative of the broader population, in part because higher education has remained unattractive or inaccessible to large segments of society (*Wapman et al., 2022*; *Davis and Fry, 2019*; *Morgan et al., 2022*).

Over the past 50 years, U.S. higher education has made substantial gains in women's representation at the undergraduate and PhD levels, but progress toward greater representation among tenure-track

faculty has been much slower. Women have earned more than 50% of bachelor's degrees since 1981 (*De Brey et al., 2021*), and now receive almost half of doctorates in the U.S. (46% in 2021) (*National Center for Science and National Science Foundation Engineering Statistics, 2021*). However, women comprise only 36% of all U.S. tenured and tenure-track faculty (*Wapman et al., 2022*), and there are significant differences in women's representation across disciplines. For example, many fewer women earn PhDs in Science, Technology, Engineering, and Mathematics (STEM) fields (38%), compared to PhD recipients in non-STEM fields (59%) (*National Center for Science and National Science Foundation Engineering Statistics, 2021*).

There are two primary ways by which women faculty representation changes: through hiring and through attrition. In our analysis of faculty demographics, faculty attrition refers to 'all-cause attrition,' which encompasses all the reasons that may lead someone to leave the professoriate, including retirement or being drawn to non-academic activities in the commercial sector. If the proportion of women among incoming hires is greater than the proportion of women among current faculty, then new hires will slightly increase the field's representation of women. On the other hand, if the proportion of women among faculty who leave their field is greater than the proportion of women among current faculty, then attrition will slightly decrease the field's representation of women. Because faculty often have very long careers, a trend in a field's overall gender representation is a cumulative integration, over many years, of the net differences in representation caused by a mixture of hiring and attrition.

Policies targeting gender parity can focus on changes to hiring, attrition, or both, and many have been tried. Hiring-focused policies include grants for diverse faculty recruitment (*Kyaw, 2022*), efforts to reduce bias in the hiring processes (*Carlson and Zorn, 2021*), and a range of other measures intended to increase women's representation at earlier educational and career stages. Attrition-focused policies include initiatives to reduce gender bias in the promotion and evaluation of faculty (*Margaret and Rommel, 2018*), efforts to diminish the gender wage gap among faculty (*Samaniego et al., 2023*), and diversity-focused grants for early and mid-career faculty research (*National Science Foundation, 2023*). Some policies simultaneously impact hiring and attrition. For example, improvements to parental leave and childcare policies may lessen the attrition of women who become parents as faculty and also encourage more prospective women faculty to consider faculty careers.

Although both types of strategy can be important, their impact alone or together on historical trends in gender diversity remain unclear. Also, we lack a clear prediction of how gender diversity may change in the future and whether current trends, and the policies that support them, may ultimately achieve gender parity. Some studies have used empirically informed models of faculty hiring, attrition, and promotion to estimate the effectiveness of certain specific policy interventions (*Marschke et al., 2007*; *Thomas et al., 2015*; *James and Brower, 2022*; *Brower and James, 2020*; *Lawrence and Chen, 2015*; *Shaw and Stanton, 2012*). However, most focus on a single institution, which tends to limit their generalizability to whole fields or to other institutions. Field-wide assessments and cross-field comparisons are necessary to provide a clear understanding of the overall patterns and their variations. Such broad comparative analyses would support evidence-based approaches to policy work and would shed new light on the causes and consequences of persistent gender inequalities among faculty.

In this study, we aim to quantify the individual and relative impacts of faculty hiring and attrition on the historical, counterfactual, and future representation of women faculty across fields and institutions. Our models and analyses are guided by a census-level dataset of faculty employment records spanning nearly all U.S.-based PhD-granting institutions, including in 111 academic fields across the Humanities, Social Sciences, Natural Sciences, Engineering, Mathematics and Computing, Education, Medicine, Health, and Business. This wide coverage allows us to quantify broad patterns and trends in both hiring and attrition, across institutions and within fields and develop model-based extrapolations under a variety of possible policy interventions.

## Results

We take three distinct approaches in our analysis of the relative importance of faculty hiring and faculty attrition for women's representation among tenure-track faculty. First, we characterize the relative contributions of hiring and attrition to changes in women's representation across a range of academic fields over 2011–2020. Second, we model a hypothetical historical scenario over this same period in which we preserve demographic trends in hiring, but we eliminate 'gendered

attrition' by assigning equal attrition rates to men and women at each career stage. Here, gendered attrition refers only to the differences in the rates in which men and women at the same career stage leave academia. It does not refer to the absolute magnitude of the rates, which increases for both men and women in the late career as faculty approach an age where retirement is common. This counterfactual model provides data-driven estimates of what different fields' gender diversity could have been, and hence provides a general estimate of the loss of diversity due to gendered attrition over this time. Finally, we use our hiring and attrition model to make quantitative projections of the potential impact of specific changes to faculty hiring and faculty attrition patterns on the future representation of women in academia, allowing us to assess the relative impact of practical or ambitious policy changes for achieving gender parity among faculty by field and in academia overall.

For these analyses, we use a census-level dataset of employment and education records for tenured and tenure-track faculty in 12,112 PhD-granting departments across 392 PhD-granting institutions in the U.S. from 2011 to 2020 (*Academic Analytics Research Center, 2021*). We organize these data into annual department-level faculty rosters. In turn, each department belongs to at least 1 of 111 academic fields (e.g. Chemistry and Sociology) and 1 of 11 high-level groupings of related fields that we call domains (e.g. Natural Sciences and Social Sciences), enabling multiple levels of analysis. This dataset was obtained under a data use agreement with the Academic Analytics Research Center (AARC), and was extensively cleaned and preprocessed to support longitudinal analyses of faculty hiring and attrition (*Wapman et al., 2022*) (see Methods for data cleaning details).

We added gender annotations to faculty using nomquamgender, an open source name-based gender classification package that is comparable in performance to the most reliable paid name-based gender classification services (*Van Buskirk et al., 2023*). Gender annotations were applied to faculty names with high cultural name-gender associations (88%) (*Van Buskirk et al., 2023*), resulting in a dataset of $n = 268,769$ unique faculty, making up 1,768,118 person-years. The methodology we use assigns only binary (woman/man) labels to faculty, even as we recognize that gender is nonbinary. This approach is a compromise due to the technical limitations of name-based gender methodologies and is not intended to reinforce a gender binary.

We define faculty hiring and faculty attrition to include all cases in which faculty join or leave a field or domain within our dataset. For example, hires include first-time tenure-track faculty, and mid-career faculty who transition from an out-of-sample institution (e.g. from a non-U.S. or non-PhD-granting institution, or from industry). Examples of faculty attritions include faculty who leave for another job in academia to an institution outside the scope of our dataset (e.g. non-U.S. or non-PhD granting), faculty who leave the tenure-track, faculty who move to another sector, and faculty who retire. Faculty who transition from one field to another are counted as an attrition from the first field and a hire into the new field. Finally, faculty who switch institutions but remain in-sample and in the same field are not counted as hires or attritions.

## Historical impacts of hiring and attrition

Our data show a clear increase in women's representation between 2011 and 2020, increasing by an average of 4.8 percentage points (pp) across fields. Trends among new hires can drive increases in women faculty's representation, and in many fields women's representation among PhD graduates has been growing for many years (*National Center for Science and Engineering Statistics, 2021*). At the same time, attrition can also drive increases in women's faculty representation if the trends run in the opposite direction, e.g., in many fields retiring faculty are more likely to be men than women (*Appendix 1—figure 8*; *Wapman et al., 2022*). However, attrition in the early- or mid-career stages may have the opposite effect if it is gendered, e.g., when women comprise a greater proportion of those leaving academia at these career stages (*Pell, 1996*; *Spoon et al., 2023*). The balance of these inflows and outflows, relative to a field's current composition, determines whether women's overall representation will increase, decrease, or hold steady over time.

To investigate the effects of hiring and attrition on changes in women's representation between 2011 and 2020, we decomposed the total change in representation into separate hiring effects and attrition effects for each of our studied fields (*Figure 1*, *Appendix 1—table 2*; see Methods). A total of 106 (95%) of 111 fields saw an increase in women's representation overall, with hiring contributing to increases in 106 of 111 fields (95%), and attrition contributing to increases in 82 of 111 fields (74%).

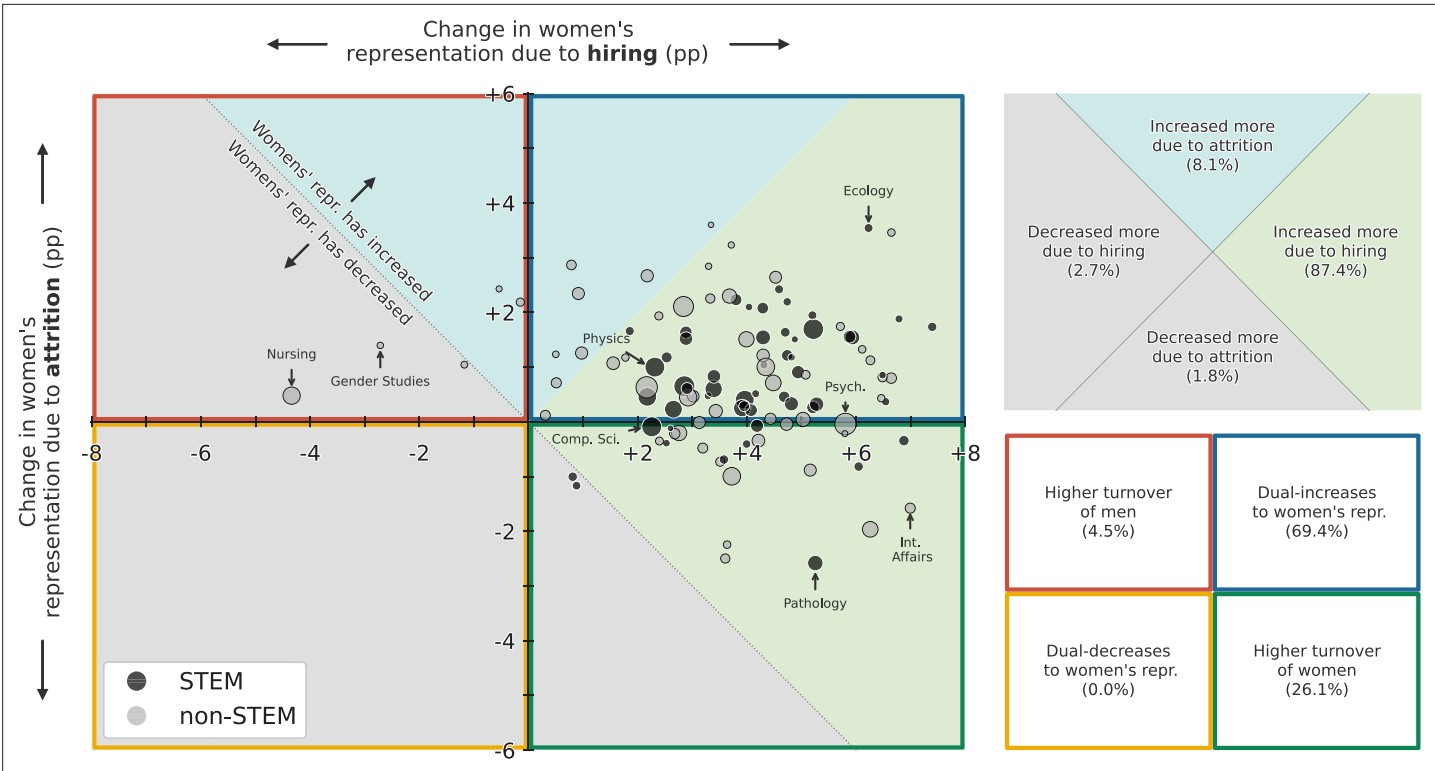

**Figure 1.** Change in women's overall faculty representation for 111 academic fields between 2011 and 2020, decomposed into change due to hiring (horizontal axis) and change due to attrition (vertical axis, see Appendix 1 section, Decomposition of change in gender diversity), showing that hiring increased women's representation for a large majority (87.4%) of fields, while it decreased women's representation for five fields. Point size represents the relative size of each field by number of faculty in 2020, and points are colored by Science, Technology, Engineering, and Mathematics (STEM) (black) or non-STEM (gray).

In general, hiring was the larger cause of increases to women's representation, with greater effects in the majority (87.4%) of academic fields.

The effects of hiring do not always increase women's representation, nor do they always dominate the effects of attrition. For instance, hiring has contributed to negative changes in women's representation in five fields, including the majority-women fields of Nursing and Gender Studies. Our analysis also finds that nine fields (8.1%) saw increases driven more by attrition than hiring, of which all are non-STEM fields.

The decomposition into hiring and attrition effects also shows that attrition can decrease women's representation, even if women's representation increases overall. In fact, faculty attrition decreased women's representation in 29 fields (26%) between 2011 and 2020 (*Figure 1*, *Appendix 1—table 2*). These attrition-driven changes likely reflect a failure to retain early and mid-career women, as they are unlikely to be driven by retirements: among faculty, men are more likely to be at or near retirement age than women faculty due to historical demographic trends (*Appendix 1—figure 8*). Indeed, of these fields, 25 (83%) of 29 are majority men, such that differential losses of women due to attrition move such fields away from parity. Nevertheless, despite net losses of women faculty to attrition, the effects of hiring were large enough to see overall increases in women's representation in 27 (93%) of 29 of these fields.

## Quantifying the impact of gendered attrition

The fact that men are systematically more likely to be at or near retirement age than women across fields (*Appendix 1—figure 8*; see also *Wapman et al., 2022*) means that it is possible for all-cause attrition to cause a net-increase in women's representation (*Figure 1*), even if women leave the academy at higher rates at every career stage. Indeed, recent work has shown that this is the case (*Spoon et al., 2023*), a phenomenon termed *gendered attrition*. These findings together suggest that

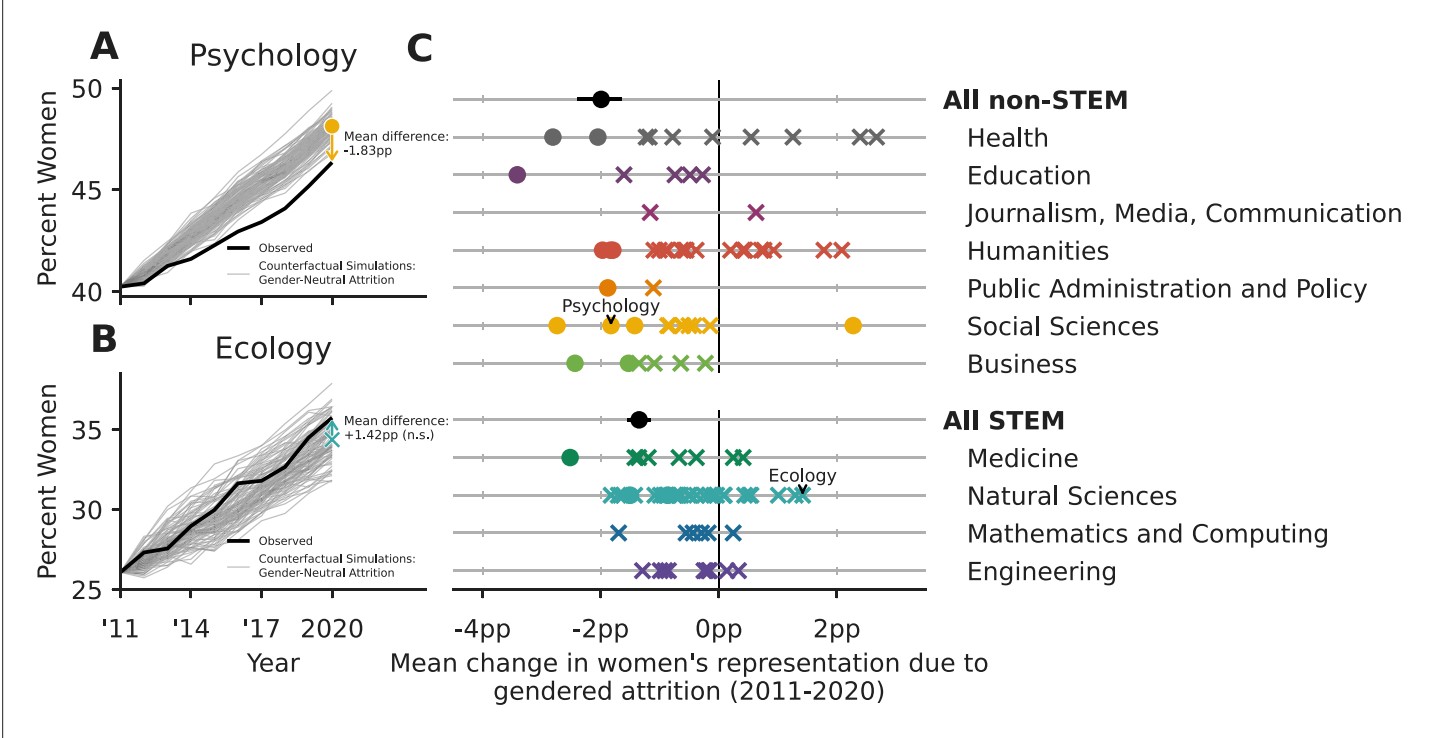

**Figure 2.** Gendered faculty attrition has caused a differential loss of women faculty in both Science, Technology, Engineering, and Mathematics (STEM) and non-STEM fields. (**A**) Gendered attrition in psychology has caused a loss of $-1.83\,\mathrm{pp}$ ($p < 0.01$) of women's representation between 2011 and 2020, relative to a counterfactual model with gender-neutral attrition (see Methods section, Model of faculty hiring an attrition). In contrast, (**B**) gendered attrition in Ecology has not caused a statistically significant loss ($+1.42\,\mathrm{pp}$, $p = 0.24$). Relative to their field-specific counterfactual simulations, 15 academic fields and the STEM and non-STEM aggregations exhibit significant losses of women faculty due to gendered attrition (circles on **C**; two-sided test for significance relative to the gender-neutral null distribution derived from simulation, $\alpha = 0.1$). The differences in the remaining 95 fields were not statistically significant (crosses on **C**), but we note that their lack of significance is likely partly attributable to their smaller sample sizes at the field-level compared to the all STEM and all non-STEM aggregations, which exhibited large and significant differences. Error bars for the non-STEM and STEM aggregations contain 95% of $n = 500$ stochastic simulations. No bars are included for field-level points to preserve readability.

in some fields, women's representation might be higher had attrition been gender neutral over the past decade.

To estimate the potential impacts of gender-neutral attrition (GNA), we created a counterfactual model in which we fixed men's and women's attrition risks to be equal at every career age (see Methods). By initializing the model at our 2011 faculty census data, and preserving demographic trends in hiring, we simulated $n = 500$ counterfactual demographic trajectories for 2011–2020 under GNA for each field. Here, our model bears an important resemblance to the seminal Leslie matrix model used and adapted by demographers and ecologists (see, e.g., *Caswell, 2001*), with a few notable differences to ensure the total faculty population size and the distribution of career ages at hiring match historical data (see Methods). From these trajectories, we quantified the effect of gendered attrition as the difference in women's representation between the real 2020 census and gender-neutral simulations, and labeled effects as significant only if 95% of simulations ended with either higher or lower representation of women. For instance, there were $1.83\,\mathrm{pp}$ fewer women in 2020 in Psychology due to a decade of significant gendered attrition (*Figure 2A*; $p < 0.01$), whereas we find no significant gendered attrition in Ecology (*Figure 2B*; $p = 0.24$).

A total of 16 fields exhibited significantly gendered attrition between 2011 and 2020 (circles, *Figure 2C*). Of these, 15 fields, including Psychology, Philosophy, Chemistry, and Sociology, ended 2020 with fewer women than our GNA model predicted, while just one ended 2020 with fewer men (Gender Studies). Counterfactual simulations for the remaining 95 fields provided inconsistent outcomes, either toward greater or lesser representation of women faculty, for at least 5% of simulations (crosses in *Figure 2C*, *Appendix 1—table 2*). In general, simulations for smaller fields tended to exhibit more variable outcomes which were consequently less often statistically significant.

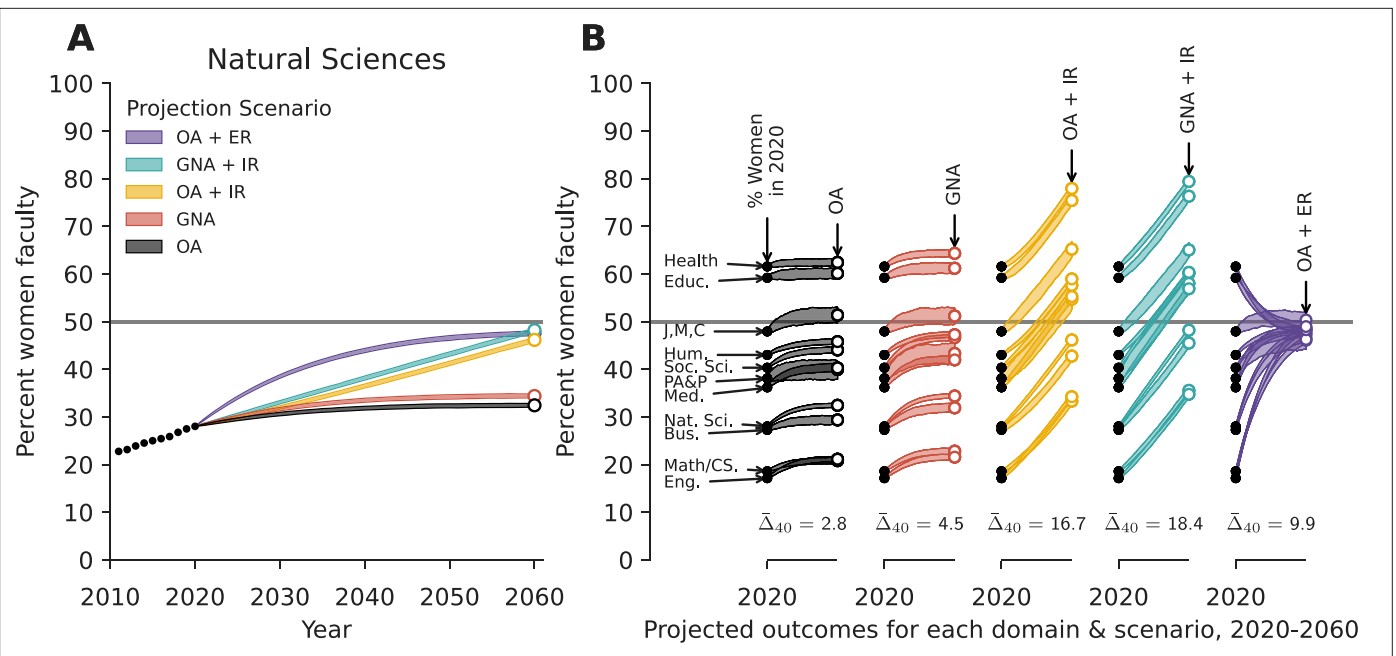

**Figure 3.** Projecting women's representation under five policy scenarios. (**A**) Observed (dotted line, 2011–2020) and projected (solid lines, 2021–2060) faculty gender diversity for Natural Sciences over time and (**B**) projections for 11 academic domains over 40 years under five policy scenarios. Line widths span the middle 95% of $N = 500$ simulations and $\bar{\Delta}_{40}$ gives the mean change in women's representation across domains over the 40-year period. Educ.=Education, J,M,C=Journalism, Media and Communications, Hum.=Humanities, Soc. Sci.=Social Sciences, and PA&P=Public Administration and Policy., Med.=Medicine, Nat. Sci.=Natural Sciences, Bus.=Business, Eng.=Engineering. See text for scenario explanations. OA = observed attrition, GNA = gender-neutral attrition, IR = increasing representation of women among hires (+0.5 pp each year), ER = equal representation of women and men among hires.

To evaluate the effects of gendered attrition at a higher level of aggregation, we also estimated the impacts of gendered attrition for all STEM and all non-STEM fields, respectively. Gendered attrition was significant in both cases, decreasing women's representation by $-1.35\,\text{pp}$ (STEM, $\text{p} < 0.01$) and $-1.99\,\text{pp}$ (non-STEM, $\text{p} < 0.01$) relative to counterfactual simulations (*Figure 2C*). Aggregating all fields, our counterfactual model estimates that gendered attrition has caused a net loss of 1378 women faculty from the PhD-granting sector of the U.S. tenure track between 2011 and 2020. Assuming 19.2 faculty per department (the mean department size in our dataset), this is an asymmetric loss of approximately 72 entire departments.

## Projecting future gender representation

Proposed strategies to increase faculty gender diversity often emphasize changes to hiring or retention. However, administrators and policymakers typically lack any ability to quantitatively evaluate a policy's long-term impact or to compare its outcomes against alternatives. In our third analysis, we therefore turn our counterfactual model, previously used to investigate the past, to investigate five projections capturing future hiring and attrition scenarios. These scenarios are not intended to predict or forecast precisely what will occur in the future, but instead serve to illustrate what could happen if a set of specific assumptions were held fixed. We operationalize a particular policy intervention by altering two parameters: faculty attrition risks, which can be gendered or gender-neutral, and the fraction of women among new hires, which can be maintained at 2011–2020 levels or increased over time. These scenarios are described in detail in Methods section, Parameters for projection model of 2020–2060. Across all scenarios, we hold the sizes of academic domains fixed at their 2020 values, initialize projections at empirical 2020 values, and project women's representation through 2060.

Even in the absence of interventions, our baseline projection using current hiring patterns and observed attrition (OA) shows a mean increase in women's representation of $2.8\,\text{pp}$ (*Figure 3*). Although the fraction of women among new hires has been increasing over time in some academic domains (Appendix 1 section, Gender diversity of hires over time), here we do not extrapolate that

trend into the future. Thus, this projection reflects demographic inertia, where it takes roughly a full career-length of time for the most recent, more gender diverse cohorts of newly hired faculty to fully replace all the older and less gender diverse faculty cohorts (*Marschke et al., 2007*; *Hargens and Long, 2002*).

The GNA scenario maintains the same assumption about hiring as the OA scenario, but alters the attrition risks to be equal for men and women at each career stage. This scenario therefore represents a set of policy interventions that entirely close the retention gap between women and men faculty. The resulting projected fraction of women faculty in the Natural Sciences domain in 2060 is 34.4%, representing a 6.4 pp increase from 2020 (*Figure 3A*). Across all domains, the mean increase in women's representation relative to 2020 is 4.5 pp (*Figure 3B*), exceeding the mean increase in the OA scenario by 1.7 pp. Nevertheless, in this scenario women are projected to remain underrepresented in most academic domains in 2060.

The increased representation (IR) scenarios alter new faculty hiring such that women's representation among new hires grows by +0.5 pp each year. This rate of growth ultimately increases women's representation among new hires by +20 pp by 2060, and is close to the median observed change 2011–2020 (+0.58 pp; *Appendix 1—table 1*). Thus, for some domains, this scenario may not represent new action, but rather continued effects of current policies.

When increased representation among new hires is combined with observed attrition (OA+IR), the projected mean increase in women's overall representation is 16.7 pp (*Figure 3*). The magnitude of this increase is the largest among these first three scenarios, exceeding the OA scenario's mean increase by 12.2 pp. We note, however, that these increases would grow women's representation beyond gender parity in Medicine, Social Sciences, Humanities, and Public Administration and Policy, and would maintain representation above parity in Health and Education (*Figure 3*).

When increased representation among new hires is instead combined with gender-neutral attrition (GNA+IR), the projected mean increase in women's overall representation is 18.4 pp (*Figure 3*), exceeding the OA+IR scenario by only 1.7 pp. These additional modest increases do not push any additional academic domains beyond gender parity (*Figure 3*).

Finally, we consider a more radical intervention in which universities immediately and henceforth hire men and women faculty at equal rates (ER) but do not change attrition patterns (OA+ER). Overall, the projected mean increase in women's representation between 2020 and 2060 is 9.9 pp (*Figure 3*), a result of all domains moving markedly toward parity. However, none of the academic domains is projected to achieve stable gender parity under this scenario because attrition remains gendered. In domains that are particularly male-dominated, such as Natural Sciences, this gender-parity hiring scenario causes the most rapid progress toward greater representation of women faculty of any of the scenarios (*Figure 3A*).

Intentionally omitted is the scenario of equal rate hiring and gender-neutral attrition (ER+GNA), for which conclusions can be drawn without simulation. Any academic domain with hiring at parity and equal retention rates between women and men is guaranteed to achieve stable parity, modulo stochastic effects, after one complete academic generation.

The five projection scenarios show that changes to hiring drive larger increases in the long-term representation of women faculty of a field. Most fields, and particularly those with the least gender diversity, cannot achieve equal representation by changing attrition patterns alone. In fact, our results suggest that even relatively modest annual changes in hiring tend to accumulate to substantial field-level changes over time. In contrast, eliminating gendered attrition leads to only modest changes in women's projected representation (*Figure 3*).

## Discussion

In this study, we used a decade of census-level employment data on U.S. tenured and tenure-track faculty at PhD-granting institutions to investigate the relative impacts of faculty hiring versus all-cause faculty attrition on women's representation across academia. Toward this end, we answer three broad questions: (i) How have these two processes shaped gender representation across the academy over the decade 2011–2020? (ii) How might women's representation today have been different if gendered attrition among faculty were eliminated in 2011? (iii) And, how might we expect gender diversity among faculty to change over time under different future hiring and attrition scenarios?

The effects of hiring were stronger than the effects of attrition in changing women's representation among faculty over the decade 2011–2020. By separating hiring and attrition effects, we found that hiring served to increase women's representation in 96% of fields, while attrition (including retirement) served to increase women's representation in 74% of fields. However, these effects were not always synergistic, such that 26% of fields saw increases due to hiring amidst decreases due to attrition, while 5% of fields saw decreases due to hiring and increases due to attrition. In total, hiring effects dominated attrition effects in 90% of fields, 97% of which saw net increases in women's representation (*Figure 1*). In contrast, attrition effects were stronger than hiring effects in just 10% of fields, yet 9 of these 11 nevertheless saw net increases in women's representation, including Linguistics, Theological Studies, and Art History. Only 5 (4.5%) fields saw decreased women's representation over same period, including Nursing and Gender Studies, both fields where women are overrepresented (*Figure 1*).

Our counterfactual analyses of the past decade's gender diversity trends, in which we preserved historical hiring trends but eliminated the effects of gendered attrition, indicate that U.S. academia as a whole has lost approximately 1378 women faculty because of gendered attrition. This number is both a small portion of academia (0.67% of the professoriate) and a staggering number of individual careers, enough to fully staff 72 nineteen-person departments. Because women are more likely to say they felt 'pushed' out of academic jobs when they leave (*Spoon et al., 2023*), each of these careers likely represents an unnecessary loss, both to the individual and to society, in the form of lost discoveries (*Hofstra et al., 2020*; *Yang et al., 2022*; *Koning et al., 2021*; *Kozlowski et al., 2022*), missing mentorship (*Lockwood, 2006*; *Stout et al., 2011*), and many other contributions that scholars make. Moreover, although our study focuses narrowly on gender, past studies of faculty attrition (*Griffin et al., 2011*; *Jayakumar et al., 2009*; *Liu et al., 2019*) lead us to expect that a disproportionate share of these lost careers would be women of color.

The two findings above—that all-cause attrition served to increase women's representation in 74% of fields, while gendered attrition led to the net loss of an estimated 1378 women from the academy—seem at first glance to be at odds. However, the former reflects primarily the turnover and retirement of a previous generation of faculty (*Wapman et al., 2022*), while the latter stems from observations that in most areas of the academy, women are at higher risk of leaving their faculty jobs than men of the same career age, i.e., gendered attrition (*Spoon et al., 2023*). Reconciling and quantifying these separate effects through models, parameterized by gender and career-age stratified empirical data, is therefore a key contribution of this work.

We find large and statistically significant effect sizes for gendered attrition at high levels of aggregation in our data, (e.g. all of academia, among all STEM or all non-STEM fields, and within domains), yet we often find smaller effects that are not statistically significant in constituent individual fields (*Figure 2*; see also *Spoon et al., 2023*). This pattern implies that there must be gendered effects within at least some of the constituent fields which may statistically indistinguishable from noise due to smaller population sizes and greater relative fluctuations in hiring. This perspective may therefore explain why field-specific studies of gendered faculty attrition sometimes reach conflicting conclusions (*Lawrence and Chen, 2015*; *Spoon et al., 2023*; *Kaminski and Geisler, 2012*; *Gumpertz et al., 2017*; *Carr et al., 2018*), and suggests that future studies should seek larger samples sizes whenever possible.

While this study cannot identify specific causal mechanisms that drive gendered attrition, the literature points to a number of possibilities, including disparities in the levels of support and value attributed to women and the scholarly work that women produce (*Macaluso et al., 2016*; *Ni et al., 2021*), sexual harassment (*National Academies of Sciences, 2018*), workplace culture (*Spoon et al., 2023*), work-life balance (*Deutsch and Yao, 2014*; *Martinez et al., 2017*), and the unequal impacts of parenthood (*Cech and Blair-Loy, 2019*; *Morgan et al., 2021*). Even in fields where we found no significant evidence that women and men leave academia at different rates, the reasons women and men leave may nevertheless be strongly gendered. For instance, past work has shown that men are more likely to leave faculty jobs due to attractive alternate opportunities ('pulls'), while women are more likely to leave due to negative workplace culture or work-life balance factors ('pushes') (*Spoon et al., 2023*).

The relative importance of hiring vs attrition is also borne out in our projection scenarios. Indeed, we find that eliminating the gendered attrition gap, in isolation, would not substantially increase

representation of women faculty in academia. Rather, progress toward gender parity depends far more heavily on increasing women's representation among new faculty hires, with the greatest change occurring if hiring is close to gender parity (*Figure 3*).

A limitation to this study is that it only considers tenured and tenure-track faculty at PhD-granting institutions in the U.S. Non-tenure-track faculty, including instructors, adjuncts, and research faculty, are an increasing portion of the professoriate (*Finley, 2009*; *McNaughtan et al., 2017*), and they may experience different trends in hiring and attrition, as may faculty outside the U.S. and faculty at institutions that do not grant PhDs. The results presented in this work will only generalize to these other populations to the extent that they share similar attrition rates, similar hiring rates, and similar current demographics. Notably, understanding how faculty hiring, faculty attrition, and faculty promotion (*Danell and Hjerm, 2013*) are shaping the gender composition of these populations is an important direction for future work.

Another important limitation of this work is that we focused our analysis at the level of entire fields and academic domains. However, hiring and attrition may play different roles in specific departments or in specific types of departments. These differences may be elucidated by future analyses of mid-career moves, in which faculty change institutions but stay in the same fields.

Faculty who belong to multiple marginalized groups (e.g. women of color) are particularly under-represented and face unique challenges in academia (*Crenshaw, 2013*). The majority of women faculty in the U.S. are white (*National Center for Education Statistics, 2022*), meaning the patterns in women's hiring and retention observed in this study are predominantly driven by this demographic. While attrition may not be the primary challenge for women's representation overall, it could still be a significant barrier for women of color and those from less privileged socioeconomic backgrounds. Additional data are needed to study trends in faculty hiring and faculty attrition across racial and socioeconomic groups.

More broadly, while this study has focused on a quantitative view of men's and women's relative representations, we note that equal representation is not equivalent to equal or fair treatment (*Tienda, 2013*; *Smith-Doerr et al., 2017*). Pursuing more diverse faculty hiring without also mitigating the causes that sustain existing inequities can act like a kind of 'bait and switch,' where new faculty are hired into environments that do not support their success, a dynamic that is believed to contribute to higher turnover rates for women faculty (*Slay et al., 2019*).

Our study's detailed and cross-disciplinary view of hiring and attrition, and their relative impacts on faculty gender diversity, highlights the importance of sustained and multifaceted efforts to increase diversity in academia. Achieving these goals will require a deeper understanding of factors that shape the demographic landscape of academia.

## Methods
### Data cleaning and preparation

Our analysis is based on a comprehensive dataset of U.S. faculty at PhD-granting institutions, obtained through a data use agreement with the AARC. This dataset includes the employment records for all tenured and tenure-track faculty at all 392 U.S. doctoral-granting universities from 2011 to 2020, along with the year of each professor's terminal degree. To ensure the consistency and robustness of our measurements, we cleaned and preprocessed this dataset according to the following steps. For additional technical details relating to these steps, see *Wapman et al., 2022*. For a manual audit to assess potential attrition errors in this dataset, see *Spoon et al., 2023*.

We first de-duplicated departments. This involved combining records due to: (i) variations in department names (e.g. 'Computer Science Department' vs. 'Department of Computer Science') and (ii) departmental renaming events (e.g. 'USC School of Engineering' vs. 'USC Viterbi School of Engineering').

Next, we annotated departments according to a two-level department taxonomy with lower level fields and higher level domains assignments to departments. While most departments were assigned a single annotation for each level of the taxonomy, some interdisciplinary departments received multiple annotations. This deliberate choice can reflect how a 'Department of Physics and Astronomy' is relevant to both 'Physics' within the 'Natural Sciences' domain and 'Astronomy' within the same

domain. Therefore, we included all applicable annotations for such departments to capture their full scope, but note that domain-level analyses included such departments only once.

Then, we addressed certain interdisciplinary fields which could conceptually reside in multiple domains, e.g., Computer Engineering (potentially belonging to domains of either Mathematics and Computing or Engineering), and Educational Psychology (potentially belonging to domains of either Education or Social Sciences). To address this ambiguity, we employed a heuristic approach. Fields were assigned to the domain containing the largest proportion of faculty members whose doctoral universities housed a department within that domain. Thus, we grouped fields based on the domain where their faculty were most likely to have been trained. *Appendix 1—table 2* contains a complete list of fields and domains.

In rare instances, faculty members temporarily disappeared from the dataset before reappearing in their original departments. We treated these as likely data errors and imputed the missing employment records. Missing records were filled in if the faculty member's department had data for the missing years. Employment records were not imputed if they were associated with a department that did not have any employment records in the given year. Imputations affected 1.3% of employment records and 4.7% of faculty.

Next, we limited our analyses to departments consistently represented in the AARC data across the study period (2011–2020). This exclusion was necessary because not all departments were consistently recorded by AARC. Departments appearing in the majority of years within the study period were retained, resulting in the removal of 1.8% of employment records, 3.4% of faculty, and 9.1% of departments. This exclusion also resulted in the removal of 24 institutions (6.1%), primarily seminaries.

We next filtered the data to include only tenure-track faculty. This involved removing temporary faculty, including non-tenure-track instructors holding titles such as 'lecturer,' 'instructor,' or 'teaching professor' at any rank, individuals with missing rank information, and faculty classified as 'research,' 'clinical,' or 'visiting.' This filtering process resulted in a dataset solely comprised of tenured and tenure-track faculty holding the titles of 'assistant professor,' 'associate professor,' and 'full professor'.

Finally, we defined career age for each person-year record in our dataset as the difference between the given year for the record and the year that the faculty member received their doctoral degree. However, doctoral year was missing for 29,872 faculty members (9.8% of faculty), necessitating their exclusion from the counterfactual analysis (Results section, Quantifying the impact of gendered attrition) and the forecasting analysis (Results section, Projecting future gender representation).

## Decomposition of changes in representation into hiring and attrition

We decomposed the annual changes in women's representation (in units of proportion per year) within each field into hiring ($\delta_{hiring}$) and attrition ($\delta_{attrition}$) components as follows. First, we define $n_w$ and $n_m$ as the counts of women and men faculty in the field in a given index year. Next, we let $h_w$ and $h_m$ be the counts of women and men that were hired between the index year and the following year, and similarly let $x_w$ and $x_m$ be the counts of women and men among faculty 'all-cause' attritions between the index year and the following year. We then approximate $\delta_{hiring}$ and $\delta_{attrition}$ as

$$\delta_{hiring} = \frac{n_w + h_w}{n_w + h_w + n_m + h_m} - \frac{n_w}{n_w + n_m}$$

$$\delta_{attrition} = \frac{n_w - x_w}{n_w - x_w + n_m - x_m} - \frac{n_w}{n_w + n_m}.$$

These equations are developed in Appendix 1 section, Decomposition of change in gender diversity. The annual changes $\delta_{hiring}$ and $\delta_{attrition}$ were summed, respectively, over 2011–2020 for each field, to construct *Figure 1*.

## Model of faculty hiring and attrition

We developed a model of annual faculty hiring and attrition structured by faculty career age (years since PhD) and gender, allowing model parameters to vary as a function of both. This model was used for both the counterfactual historical analysis, which investigated gendered attrition, and the projection of future scenarios, which investigated five stylized futures. Details of the particular parameterizations for each investigation follow this structural model description.

In this model, we track the number of people with a given career age $a$ and gender $g$, with annual updates given by

$$n(a, g) \leftarrow n(a - 1, g) + h(a, g) - x(a - 1, g), \tag{1}$$

where $h$ and $x$ are hires and attritions, respectively. In each stochastic model realization, both $h$ and $x$ are drawn according to distributions that allow control over the extent to which hiring and attrition are (or are not) gendered processes.

We stochastically draw $h(a, g)$ and $h(a, \tilde{g})$ by first specifying the total number of hires that year $H(a)$. For each of the $H(a)$ hires, we draw gender annotations from independent Bernoulli trials with parameter $\psi(a, g)$. In this way, the $\psi$ parameters control the extent to which hiring is gendered across career ages.

To calculate attritions $x(a - 1, g)$, we subject each of the $n(a - 1, g)$ sitting faculty to a career age and gender-stratified annual attrition risk $\phi(a - 1, g)$, realizing the actual number of attritions from a binomial draw with $n$ trials and parameter $\phi$. The relative values of $\phi(a, g)$ and $\phi(a, g)$ therefore control the extent to which attrition is gendered for a particular career age.

Thus, this model is stochastic, and after specifying initial conditions for the values of $\mathbf{n}$, one needs only values for the total hires $\mathbf{H}$, gendered hiring parameters $\psi$, and gendered attrition risk parameters $\phi$ to simulate stochastically, by iterating *Equation 1*.

## Parameters for counterfactual model of 2011–2020

The goal of this model was to quantify the impact of gendered attrition over the period 2011–2020. As such, this model manipulated the parameters $\phi$ which control the extent to which attrition is gendered. To model a counterfactual scenario in which attrition was not gendered, we set women's and men's attrition risks to be identical $\phi(a, w) = \phi(a, m)$, taking on values estimated from empirical data in which gender was ignored, for each field (see below).

For each field, the model's faculty counts $\mathbf{n}$ were initialized using our 2011 faculty roster data, thereby matching all empirical 2011 values for both gender and career age. To ensure that a field's total faculty size grew or shrunk each year in a manner that exactly matched empirical changes, we first drew all attrition values $x$ and then set the total number of new hires accordingly. Those new hires were assigned initial career ages drawn from the empirical age distribution of new hires, averaged over 2012–2020. Gendered hiring parameters $\psi$ were set to values estimated from empirical data for each field (see below).

To parameterize $\psi(a, w, t)$, the probability that a new hire of career age $a$ in year $t$ is a woman, we used a logistic regression model fit to empirical hiring data for each field. Because the probability that a new hire is a woman varies non-linearly with career age (see *Appendix 1—figure 4*), the dependent variables in this model include new hires' career ages up to their fifth exponents and a linear term for the calendar year, which ranges from 2012 to 2020.

To parameterize $\phi(a, g, t)$, the probability that faculty of career age $a$ and gender $g$ experience attrition in year $t$, we used a logistic regression model trained on empirical attrition and retention data for each field. Because faculty attrition rates vary non-linearly with career age (see *Appendix 1—figure 3*), we include new hires' career ages up to their fifth exponents as dependent variables in the regression model, in addition to a linear term for the year, which ranges from 2012 to 2020. This regression model is fit to all observed cases of attrition and retention for both men and women together, for each field. Accordingly, men and women in our gender-neutral counterfactual model were subjected to the same age-varying attrition risks, eliminating the gendered aspect of these patterns, while preserving the rises and declines in attrition rates across faculty career ages.

To capture the distribution of counterfactual historical outcomes under gender neutral attrition, we drew 500 simulations of the years 2012–2020 for each field, recording women's representation at the end of each. See *Appendix 1—figure 2* for details of model validation. This model was run for each field, independently. It was also run again for all STEM fields, and all non-STEM fields, with respective parameters estimated at those respective higher levels of data aggregation (*Figure 2*).

## Parameters for projection model of 2020–2060

The goal of this model was to quantify the effects of a set of five highly stylized scenarios for how hiring and attrition might evolve over the 40 years spanning 2020–2060. For scenarios with GNA, we

set women's and men's attrition risks to be identical $\phi(a, w) = \phi(a, m)$, taking on values estimated from empirical data in which gender was ignored, for each academic domain. For scenarios with OA, we set women's and men's attrition risks to values estimated from empirical data in which gender was included, for each domain. For scenarios with equal representation of women and men among hires (ER), we set all $\psi$ parameters to 0.5. And, for scenarios with increasing representation of women among hires (IR), we let the hiring parameters vary over time, such that the expected proportion of women among new hires increases by 0.5 pp per year for each career age starting in 2020, i.e.,

$$\psi(a, w, t) = \psi(a, w, 2020) + 0.005(t - 2020).$$

In the absence of one of the above manipulations, $\phi$ and $\psi$ parameters were set to their empirical values, estimated from aggregated 2011–2020 data for each domain.

For each academic domain, the model's faculty counts $\mathbf{n}$ were initialized using our 2020 faculty roster data, thereby matching all empirical 2020 values for both gender and career age. Each domain's total faculty size was held fixed at 2020 values by setting the total number of new hires to be equal to the number of stochastically drawn attritions. This model was run for each domain, independently, using parameters estimated at the domain level accordingly.

To capture the distribution of outcomes in each projection scenario, we drew 500 simulations of the years 2020–2060 for each academic domain, recording women's representation at the end of each (*Figure 3*). See Appendix 1 section, Model validation and sensitivity analysis, for details of model validation.

## Acknowledgements

We thank K Spoon, I Van Buskirk, and B Fosdick for helpful comments, and the AARC for providing the data that made these analyses possible. Air Force Office of Scientific Research Award FA9550-19-1-0329 (NL, KHW, AC, DBL), National Science Foundation Alan T Waterman Award SMA-2226343 (DBL). The funders had no role in study design, data collection and analysis, decision to publish or preparation of the manuscript.

## Additional information

### Funding

| Funder | Grant reference number | Author |
|---|---|---|
| Air Force Office of Scientific Research | FA9550-19-1-0329 | Nicholas LaBerge<br>Kenneth Hunter Wapman<br>Aaron Clauset<br>Daniel B Larremore |
| National Science Foundation | SMA-2226343 | Daniel B Larremore |

The funders had no role in study design, data collection and interpretation, or the decision to submit the work for publication.

### Author contributions

Nicholas LaBerge, Conceptualization, Data curation, Software, Formal analysis, Validation, Investigation, Visualization, Methodology, Writing – original draft, Writing – review and editing; Kenneth Hunter Wapman, Data curation; Aaron Clauset, Daniel B Larremore, Conceptualization, Resources, Supervision, Funding acquisition, Investigation, Methodology, Writing – original draft, Project administration, Writing – review and editing

### Author ORCIDs

Nicholas LaBerge http://orcid.org/0000-0002-4922-3358
Aaron Clauset http://orcid.org/0000-0002-3529-8746
Daniel B Larremore https://orcid.org/0000-0001-5273-5234

Reviewer #1 (Public Review): https://doi.org/10.7554/eLife.93755.3.sa1
Reviewer #3 (Public Review): https://doi.org/10.7554/eLife.93755.3.sa2
Author response https://doi.org/10.7554/eLife.93755.3.sa3

## Additional files

### Supplementary files
• MDAR checklist

### Data availability

The dataset used for these analyses were obtained under a data use agreement with the Academic Analytics Research Center (AARC). According to this data use agreement, we cannot publicly disseminate this dataset. Interested researchers may apply directly to the Academic Analytics Research center for data access on AARC's website, here: https://aarcresearch.com/access-our-data. Although our data use agreement forbids us from sharing the microdata used to produce these analyses, we have shared the processed data that were used to produce Figure 1 and Figure 2 in Appendix 1—table 2, and we provide additional aggregate data in Appendix 1—table 3.

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

## Appendix 1

### Decomposition of change in gender diversity

We develop a method to decompose the change in each field's gender diversity into its two main components: change due to hiring and change due to attrition. First, the fraction of faculty in a field that are women in any given year can be written as

$$\frac{n_w}{n_w + n_m}$$

where $n_w$ and $n_m$ are the number of women and men faculty in the field, respectively. An additional term could be added to account for nonbinary faculty representation if our data included nonbinary gender annotations, however the methodology we use (*Van Buskirk et al., 2023*) assigns only binary (woman/man) labels to faculty.

Then, in the following year, the new fraction of women faculty can be written as

$$\frac{n_w + h_w - x_w}{n_w + h_w - x_w + n_m + h_m - x_m}$$

where $h_w$ and $h_m$ are the numbers of women and men that were hired in the following year, respectively, and $x_w$ and $x_m$ are the number of women and men among faculty 'all-cause' attritions (whether for retirement, or otherwise). The total change in women's representation between two academic years $\delta_{total}$ can thus be written as

$$\delta_{total} = \frac{n_w + h_w - x_w}{n_w + h_w - x_w + n_m + h_m - x_m} - \frac{n_w}{n_w + n_m}$$

We decompose this total change into change due to hiring, $\delta_{hiring}$, and change due to attrition, $\delta_{attrition}$, as follows:

$$\delta_{hiring} = \frac{n_w + h_w}{n_w + h_w + n_m + h_m} - \frac{n_w}{n_w + n_m}$$

$$\delta_{attrition} = \frac{n_w - x_w}{n_w - x_w + n_m - x_m} - \frac{n_w}{n_w + n_m}$$

This decomposition behaves intuitively. For example, if the share of women that are hired exceeds the fraction of women in the field prior to hiring, then $\delta_{hiring}$ will be positive. On the other hand, if the share of women that are hired is lower than the fraction of women in the field prior to hiring, then $\delta_{hiring}$ will be negative. If there are no hires, $\delta_{hiring} = 0$. Similar intuition can be applied for $\delta_{attrition}$.

We sum the change in representation due to hiring and attrition over each year between 2011 and 2020 to get the overall change in representation due to hiring, $\Delta_{hiring}$, and attrition, $\Delta_{attrition}$.

One potential limitation of this decomposition is that $\Delta_{hiring}$ and $\Delta_{attrition}$ do not perfectly sum to the exact observed change in women's representation over a given range of years, $\Delta_{total}$. Instead, there is a residual term

$$\Delta_{residual} = \Delta_{total} - \Delta_{hiring} - \Delta_{attrition}$$

Intuitively, we know that there should be a residual term, because the change in representation that results from a given cohort of new hires can depend upon the number of attritions observed in that year, and vice versa. If the residual terms are large, this decomposition would not be a good approximation of the total change in representation, and *Figure 1* could be misleading. In *Appendix 1—figure 1* we show that the residual terms are small (ranging from –0.51 pp to 1.14 pp, median = 0.2 pp), and thus the decomposition is a good approximation of the total change in women's representation.

### Model validation and sensitivity analysis

One way that we validate this model of faculty hiring and attrition is by starting the model in 2011, and comparing the resulting gender composition of faculty with the observed gender composition of faculty in 2020. In this validation, we use the same set of model parameters as in the gender-neutral counterfactual analysis (Results section, Quantifying the impact of gendered attrition), except

attrition patterns are inferred separately for women faculty and men faculty. In other words, the attrition probabilities in this validation are not gender-neutral. We find that the observed outcomes are statistically indistinguishable from the model-based outcomes for all 111 fields, and for STEM and non-STEM aggregations (*Appendix 1—figure 2*). This finding is not surprising, because the model is fit to the observed data, but it serves to validate the methods that we used to set the model's parameters (e.g. fitting logistic regression models to infer attrition risks and to infer the fraction of women faculty among new hires, as described in Methods section, Parameters for counterfactual model of 2011–2020).

We additionally validate the model by comparing the projected 2060 faculty career age distributions for Natural Sciences from *Figure 3* with the observed career age distribution for Natural Sciences in 2020 (*Appendix 1—figure 3*). We find that the projected 2060 career age distributions are similar to the observed 2020 career age distribution for Natural Sciences (shown in *Appendix 1—figure 3*) and for the additional academic domains.

We perform a sensitivity analysis to test the robustness of our results to changes in the model's parameters. In particular, we test the robustness of our counterfactual results to different models of attrition risks and to different models of the fraction of women among new hires. First, we fit several alternative models to the empirical attrition risks (*Appendix 1—figure 4*) and to the fractions of women among new hires (*Appendix 1—figure 5*) to validate our choice of including career age up to its fifth power as a predictor in these logistic regression models. Then, we test the robustness of our counterfactual results to changes in the model's parameters by running the counterfactual analysis with the alternative model in which we only include career age up to its third power. This alternate model is less likely to overfit the data, and even tends to underfit the observed correlation structures between attrition risk and career age (*Appendix 1—figures 4 and 5*). Nevertheless, we find that the results are robust to these changes (*Appendix 1—figure 6*).

## Gender diversity of hires over time

In academia overall, the fraction of women faculty among hires has been increasing on average over the past decade, at a rate of around 0.91 pp/year (*Appendix 1—figure 7*), however, these rates of change are not uniform across academic domains. *Appendix 1—table 1* shows regression results for trends in women's representation among hires for 11 academic domains. While women's representation has been increasing in 6 of the 11 domains over time at rates up to 1.30 pp/year, the remaining 5 domains have not exhibited significant trends (*Appendix 1—table 1*).

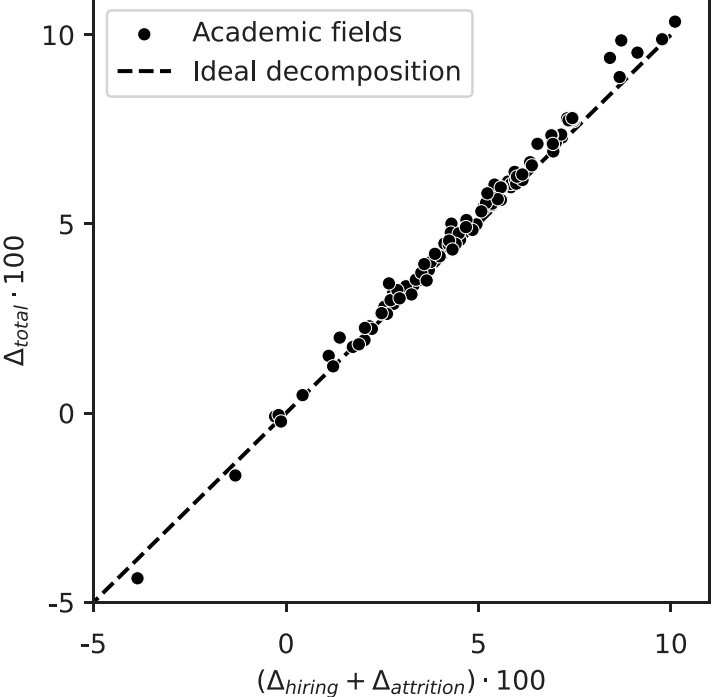

**Appendix 1—figure 1.** The change in gender diversity between 2011 and 2020 can be approximately decomposed into parts due to hiring and attrition for each academic field, but there is a leftover residual term. In practice, we find that the residual term tends to be very small, such that the decomposition is nearly ideal. The dotted line represents an ideal decomposition, where the change in women's representation among faculty due to hiring and attrition perfectly matches the total observed change.

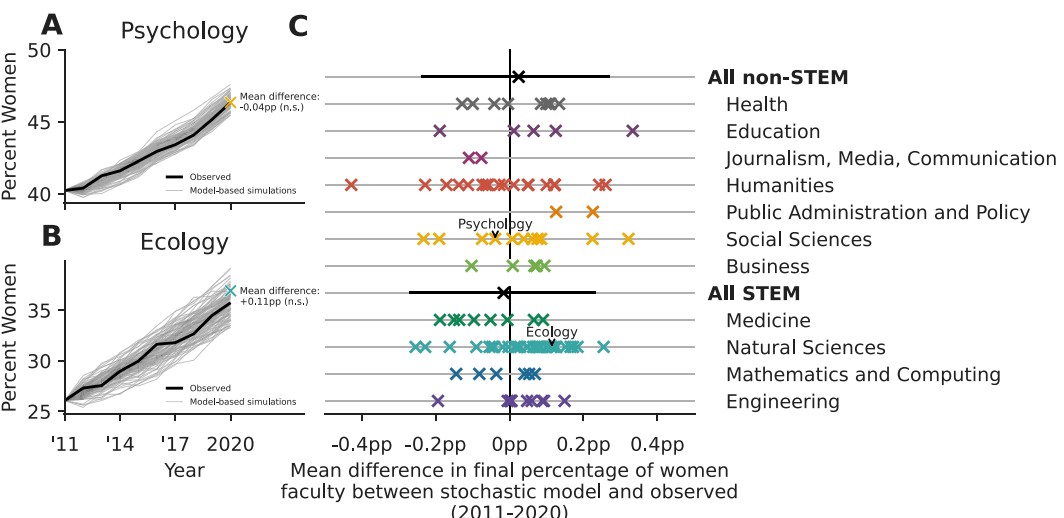

**Appendix 1—figure 2.** Model validation: Differences between observed gender diversity outcomes and model-based outcomes. (**A**) The mean outcomes of model-based simulations in psychology differ from the observed outcomes by $-0.04\,\text{pp}$, and (**B**) in Ecology by $+0.11\,\text{pp}$, but these differences are not statistically significant. (**C**) Gender diversity outcomes from model-based simulations of hiring and attrition are statistically indistinguishable from observed gender diversity outcomes for all 111 fields, and for Science, Technology, Engineering, and Mathematics (STEM) and non-STEM aggregations, based on a two-sided test for significance relative to the model-based null distribution derived from simulation, $\alpha = 0.1$. Error bars for the non-STEM and STEM aggregations contain 95% of stochastic simulations. No bars are included for field-level points to preserve readability.

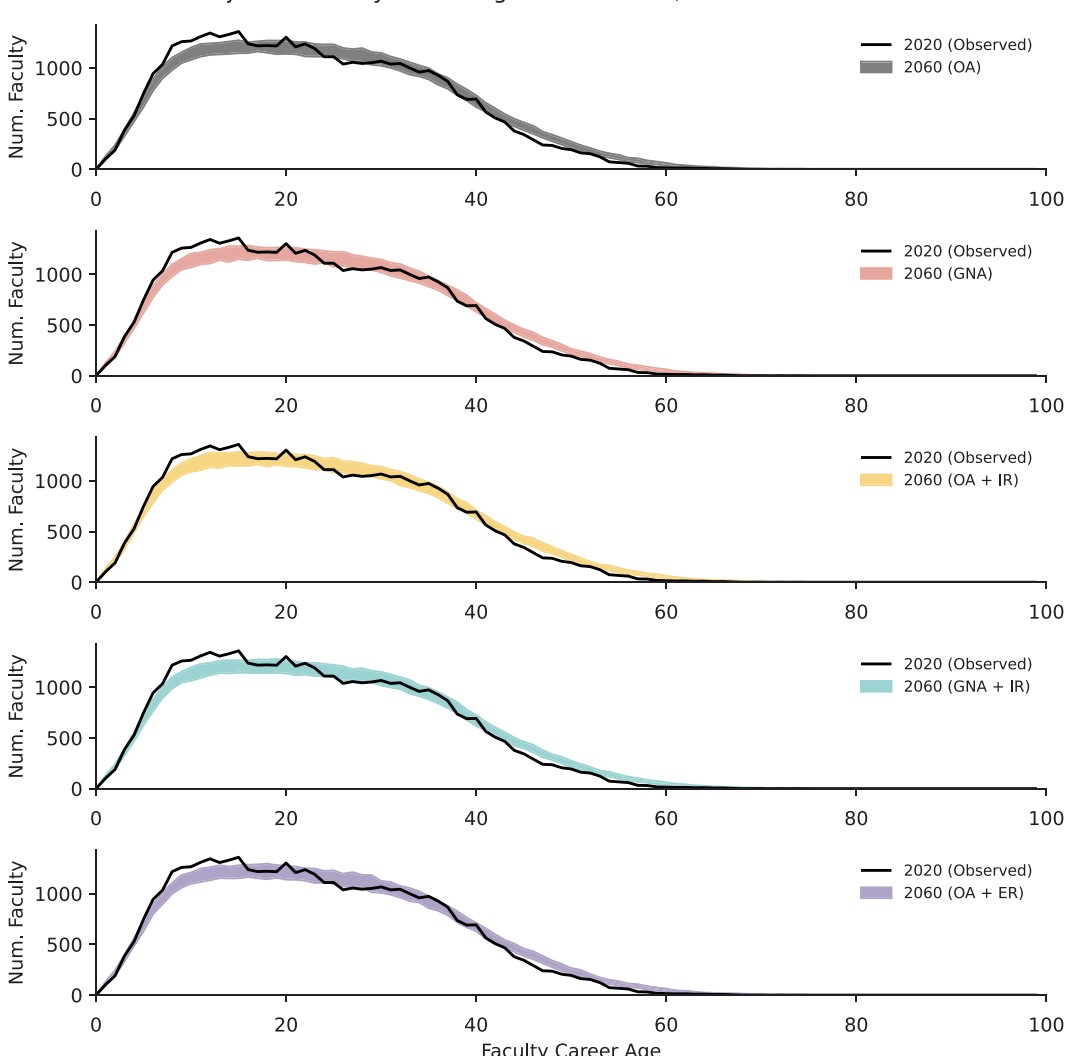

**Appendix 1—figure 3.** Model validation: Projected 2060 faculty career age distributions for Natural Sciences from **Figure 3** are similar to the observed career age distribution for Natural Sciences in 2020, for each projection scenario. Line widths for the simulated scenarios span the middle 95% of simulations. OA = observed attrition, GNA = gender-neutral attrition, IR = increasing representation of women among hires (+0.5 pp each year), ER = equal representation of women and men among hires.

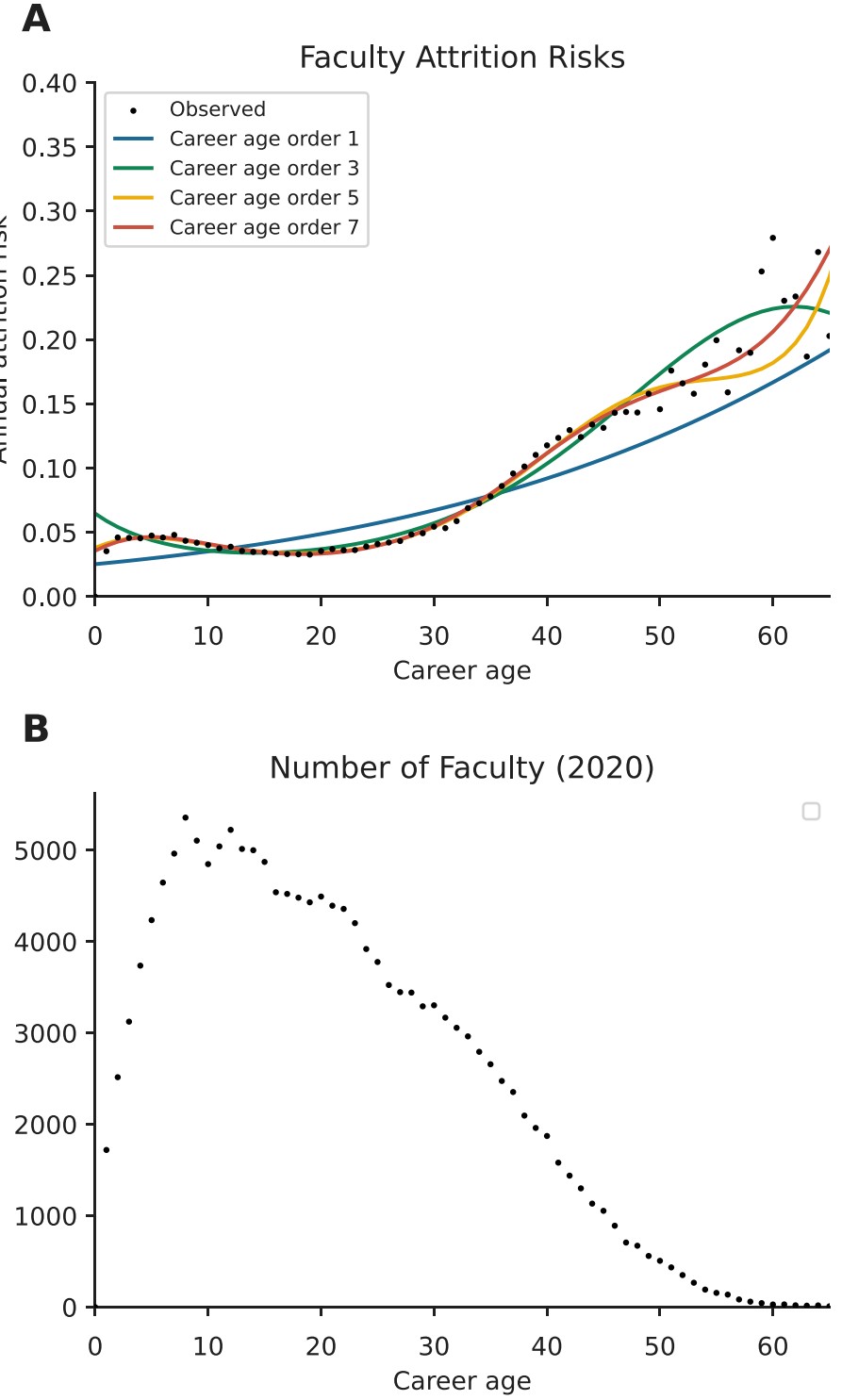

**Appendix 1—figure 4.** Model selection. (**A**) Four logistic regression models fit to observed faculty attrition data. Each model includes career age up to a different power, e.g., the model labeled 'Career age order 3' includes career age up to its third power: $logit\,(p) = \beta_0 + \beta_1 a + \beta_2 a^2 + \beta_3 a^3 + \beta_4 t$ where $a$ represents career age and $t$ represents year (see Methods section, Parameters for counterfactual model of 2011–2020, for details). The pattern in observed attrition risk becomes more noisy at higher career ages, because (**B**) there are relatively low numbers of faculty at the highest observed career ages.

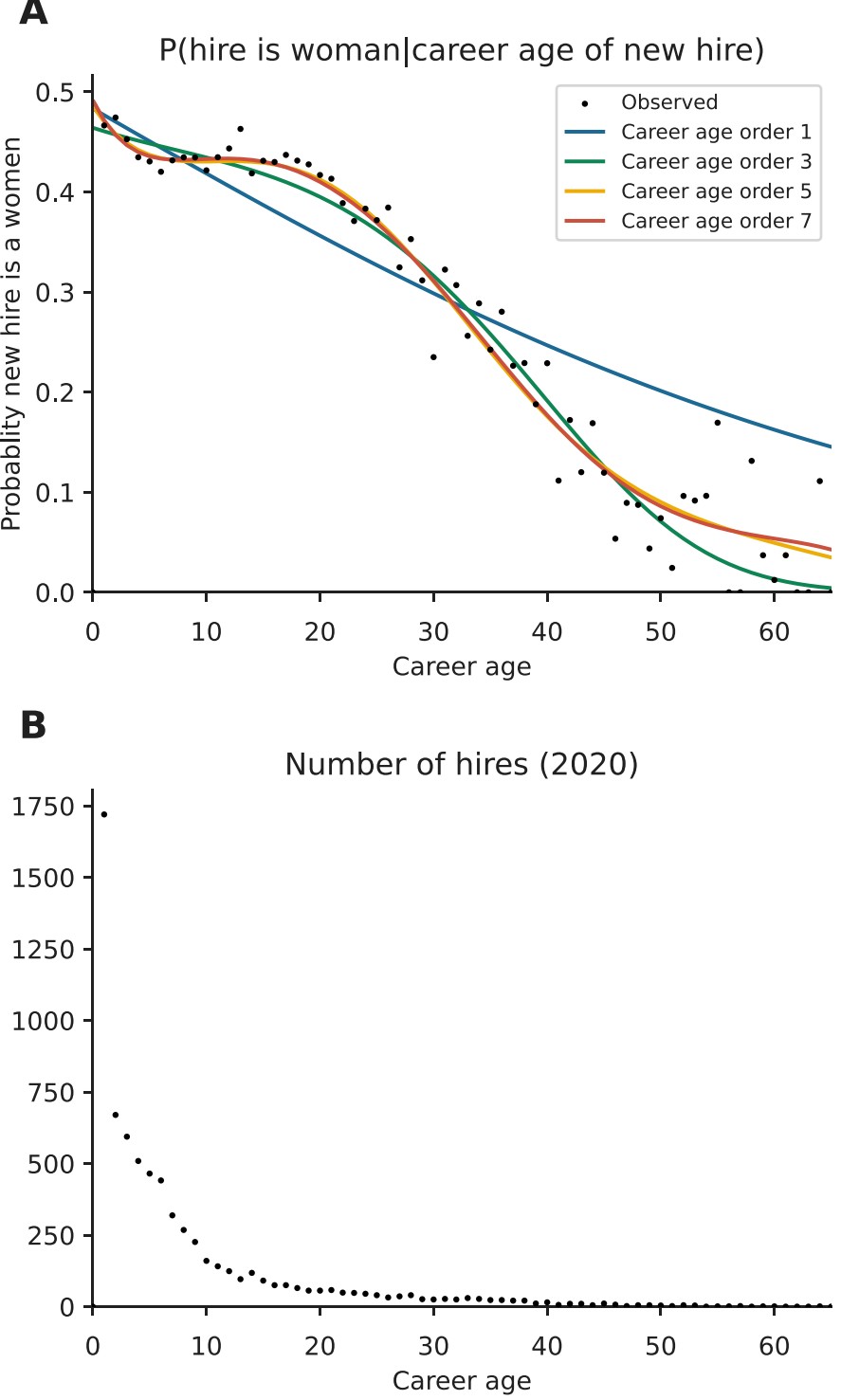

**Appendix 1—figure 5.** Model selection. (**A**) Four logistic regression models fit to observed faculty hiring data, where the outcome variable is the gender of the faculty hire (1=woman, 0=man). Each model includes career age up to a different power, e.g., the model labeled 'Career age order 3' includes career age up to its third power: $logit\left(p\right) = \beta_0 + \beta_1 a + \beta_2 a^2 + \beta_3 a^3 + \beta_6 t$ where $a$ represents career age and $t$ represents year (see Methods section, Parameters for counterfactual model of 2011–2020, for details). The pattern in the gender representation among new faculty hires becomes more noisy at higher career ages, because (**B**) there are relatively low numbers of faculty hired at higher career ages.

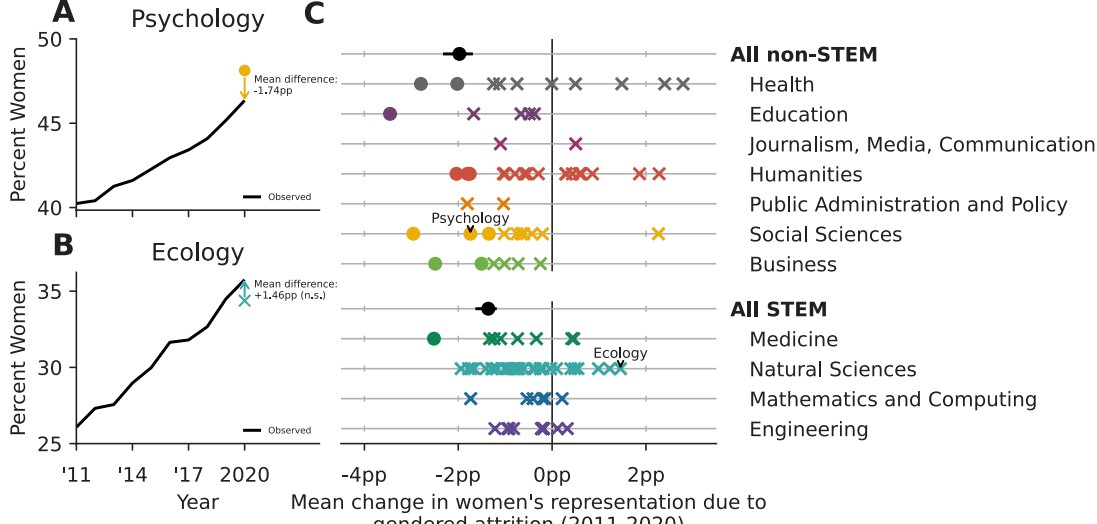

**Appendix 1—figure 6.** Sensitivity analysis: Replicating the counterfactual analysis from Results section, Quantifying the impact of gendered attrition using career age up to its third power in the associated logistic regressions model, instead of the fifth power (see Appendix 1 section, Model validation and sensitivity analysis, for details). Findings under this parameterization are qualitatively very similar to those presented in *Figure 2*, indicating that the results are robust to modest changes to model parameterization.

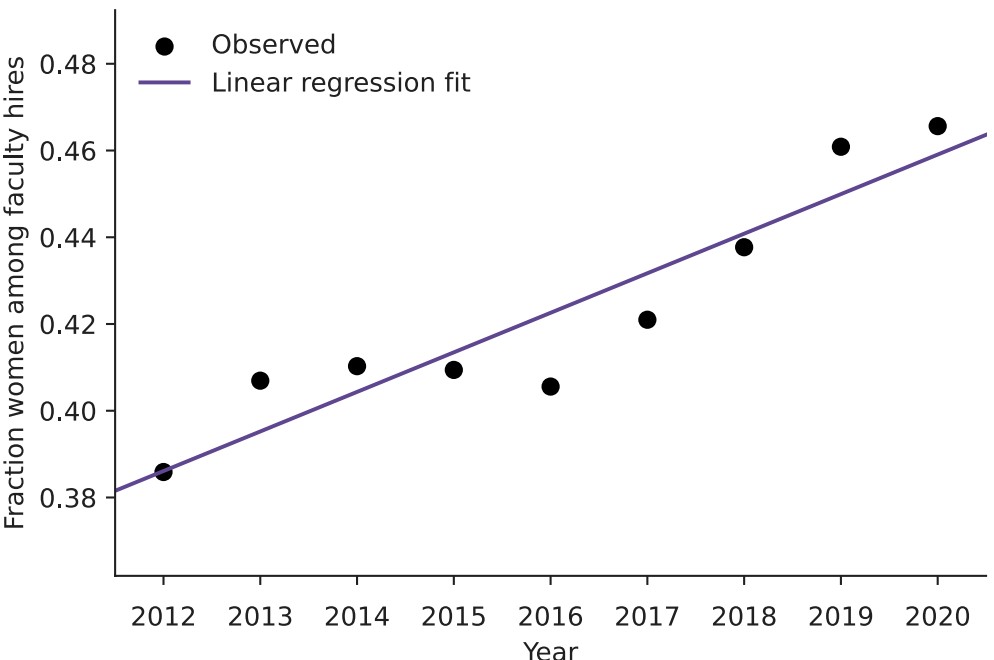

**Appendix 1—figure 7.** Fraction of women among tenure-track faculty hires over time at U.S. PhD-granting institutions. Women's share of new hires is observed to increase at around $0.91$ pp annually (t-test, $p < 0.001$), measured by an ordinary least squares regression fit (shown in purple).

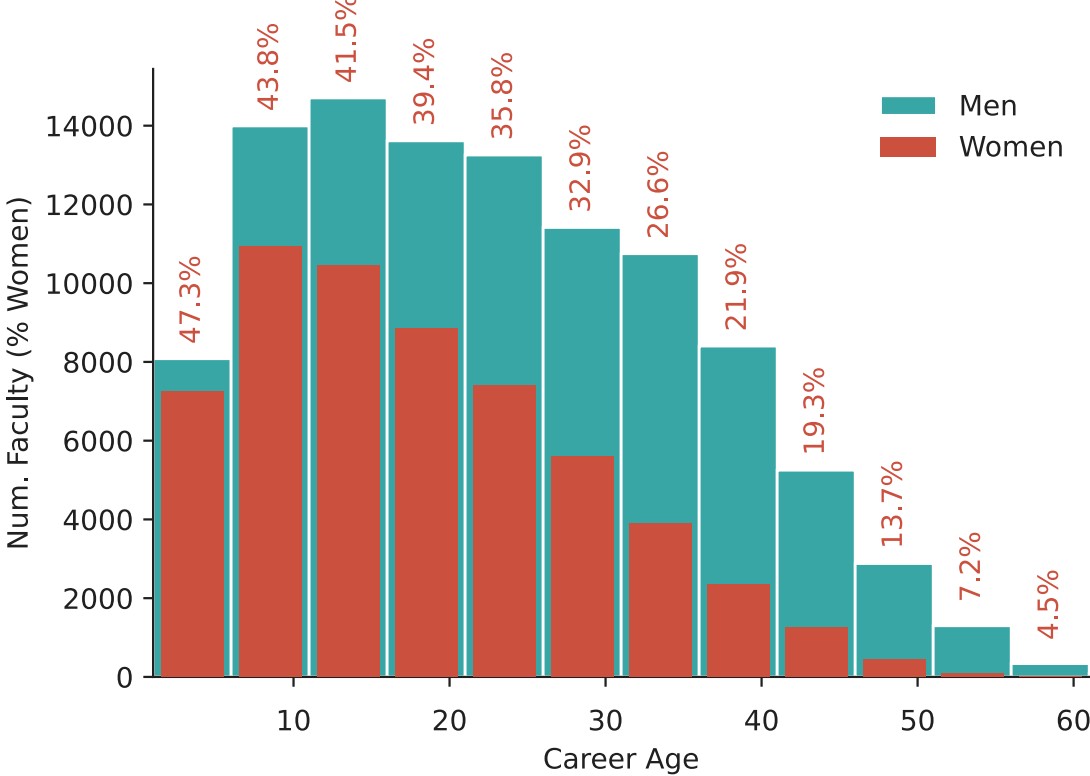

**Appendix 1—figure 8.** Career age distribution of women (red) and men (blue) tenured and tenure-track faculty across all academic fields. Career age is measured as the number of years since earning a PhD. There are substantially more men faculty with high career ages than women faculty.

**Appendix 1—table 1.** Trends in women's representation among new hires from 2012 to 2020 for 11 academic domains, along with academia overall.

We use linear regression to measure the expected change in women's concentration among new hires each year, and find that women's representation has been increasing in 6 of the 11 domains over time, at rates ranging from 0.58 pp to 1.30 pp per year. The remaining 5 domains have not exhibited significant linear trends. Overall, the fraction of women among hires has been increasing in academia over time (*Appendix 1—figure 7*). These findings are qualitatively replicated using logistic regression, so we present the linear regression results here for enhanced interpretability.

| Academic domain | Estimated annual change (pp) | p |
|---|---|---|
| Mathematics and Computing | 0.25 | 0.095 |
| Social Sciences | 1.18[†] | 0.006 |
| Natural Sciences | 1.30[‡] | 0.000 |
| Engineering | 0.95[‡] | 0.000 |
| Health | 0.58[*] | 0.020 |
| Humanities | 1.18[†] | 0.002 |
| Public Administration and Policy | 0.22 | 0.717 |
| Business | 0.38 | 0.089 |
| Medicine | 0.98[†] | 0.009 |
| Journalism, Media and Communication | 0.49 | 0.173 |
| Education | 0.34 | 0.057 |

*Appendix 1—table 1 Continued on next page*

*Appendix 1—table 1 Continued*

| Academic domain | Estimated annual change (pp) | p |
|---|---|---|
| Academia Overall | 0.91‡ | 0.000 |

*p<0.05.
†p<0.01.
‡p<0.001.

**Appendix 1—table 2.** Changes in women's representation through Hiring, Attrition, and Gendered Attrition in Academic Fields (2011–2020).
Observed changes in women's representation resulting from hiring and attrition, expressed in percentage points (pp), based on data from *Figure 1*, and the estimated average change in women's representation due to gendered attrition as depicted in *Figure 2*, accompanied by the 2.5 percentile and 97.5 percentiles of simulations in parentheses. The analysis covers 111 academic fields.

| Field | Observed $\Delta_{hiring}$ | Observed $\Delta_{attrition}$ | Est. $\Delta_{gendered\ attrition}$ |
|---|---|---|---|
| Health | | | |
| Environmental Health Sciences | +6.65 | +3.46 | +2.40 (–1.03, +5.60) |
| Nursing | –4.34 | +0.48 | +0.55 (–0.59, +1.65) |
| Public Health | +4.48 | +0.71 | –0.11 (–1.89, +1.60) |
| Human Development and Family Sciences | +3.19 | –0.48 | –2.81 (–5.44, –0.25) |
| Speech and Hearing Sciences | +3.30 | +2.84 | +2.67 (–0.81, +6.31) |
| Exercise Science, Kinesiology, Rehab, Health | +6.26 | –1.96 | –2.05 (–3.63, –0.27) |
| Nutrition Sciences | +0.31 | +0.12 | –1.18 (–3.62, +1.37) |
| Communication Disorders and Sciences | +6.11 | +1.32 | +1.26 (–1.77, +4.99) |
| Health, Physical Education, Recreation | +2.40 | –0.35 | –0.78 (–4.34, +2.52) |
| Social Work | +4.73 | –0.04 | –1.23 (3.21, +0.60) |
| Education | | | |
| Education | +1.55 | +1.07 | –0.28 (–2.21, +1.62) |
| Special Education | +3.64 | –2.24 | –3.41 (–3.41, –0.28) |
| Education Administration | +0.91 | +2.34 | –0.74 (–2.84, +1.51) |
| Counselor Education | +5.08 | +0.86 | –0.49 (–3.36, +2.49) |
| Curriculum and Instruction | +0.97 | +1.26 | –1.61 (–3.70, +0.22) |
| Journalism, Media and Communication | | | |
| Communication | +3.43 | +0.20 | –1.16 (–3.01, +0.80) |
| Mass Communications and Media Studies | +4.30 | +1.22 | +0.63 (–1.61, +2.73) |
| Humanities | | | |
| Theological Studies | +0.51 | +0.71 | –0.91 (–3.07, +1.02) |
| Asian Languages | –0.54 | +2.43 | +1.78 (–1.91, +5.32) |
| Slavic Languages and Literatures | +3.34 | +3.60 | +0.93 (–3.00, +5.18) |
| Classics and Classical Languages | +3.33 | +2.25 | +0.20 (–2.14, +2.69) |
| French Language and Literature | +0.50 | +1.23 | –0.38 (–3.69, +3.30) |
| Germanic Languages and Literatures | +3.72 | _3.23 | +2.08 (–1.44, +5.76) |
| Theatre Literature, History and Criticism | +0.79 | +2.87 | +0.40 (–4.04, +4.58) |

*Appendix 1—table 2 Continued on next page*

*Appendix 1—table 2 Continued*

| Field | Observed $\Delta_{hiring}$ | Observed $\Delta_{attrition}$ | Est. $\Delta_{gendered\ attrition}$ |
|---|---|---|---|
| Art History and Criticism | +2.18 | +2.67 | +0.44 (–1.85, +2.86) |
| Asian Studies | –1.17 | +1.04 | +0.72 (–2.84, +3.90) |
| History | +2.84 | +2.11 | –0.64 (–1.69, +0.44) |
| Urban and Regional Planning | +5.71 | +1.74 | –1.01 (–4.53, +2.34) |
| Linguistics | –0.15 | +2.19 | +0.44 (–2.12, +3.29) |
| English Language and Literature | +2.17 | +0.63 | –1.80 (–2.80, –0.80) |
| Near and Middle Eastern Languages and Cultures | +5.80 | –0.22 | –0.87 (–4.87, +3.41) |
| Music | +3.72 | –0.99 | –1.10 (–2.54, +2.54) |
| Philosophy | +5.03 | +0.04 | –1.84 (–3.37, –0.24) |
| Religious Studies | +2.68 | –0.22 | –1.97 (–4.28, +0.08) |
| Comparative Literature | +1.78 | +1.17 | –0.58 (–4.04, +2.40) |
| Spanish Language and Literature | +2.39 | +1.94 | –0.55 (–3.09, +1.98) |
| Architecture | +4.53 | +2.64 | +0.77 (–2.31, +3.72) |
| **Public Administration and Policy** | | | |
| Public Policy | +5.16 | –0.88 | –1.88 (–3.85, +0.23) |
| Public Administration | +6.49 | +0.82 | –1.11 (–3.71, +1.62) |
| **Social Sciences** | | | |
| Sociology | +3.99 | +1.51 | –1.43 (–2.92, +0.03) |
| Gender Studies | –2.71 | +1.40 | +2.28 (–0.56, +5.25) |
| Anthropology | +3.68 | +2.30 | –0.42 (–1.94, +1.31) |
| Political Science | +4.35 | +1.00 | –0.15 (–1.41, +0.98) |
| International Affairs | +6.99 | –1.58 | –2.74 (–5.31, –0.33) |
| Geography | +6.65 | +0.80 | –0.51 (–2.26, +1.82) |
| Psychology | +5.81 | –0.04 | –1.83 (–2.82, –0.74) |
| Agricultural Economics | +6.46 | +0.43 | –0.87 (–4.10, +1.76) |
| Educational Psychology | +3.51 | –0.73 | –0.84 (–3.75, +2.13) |
| Economics | +2.93 | +0.45 | –0.61 (–1.66, +0.50) |
| Criminal Justice and Criminology | +6.26 | +1.12 | –0.63 (–3.47, +2.28) |
| **Business** | | | |
| Accounting | +3.02 | +0.47 | –1.35 (–3.37, +0.61) |
| Marketing | +4.43 | +0.06 | –1.09 (–3.11, +0.99) |
| Management Information Systems | +3.61 | –2.50 | –2.44 (–4.61, –0.10) |
| Finance | +3.13 | –0.01 | –0.64 (–2.26, +0.88) |
| Business Administration | +4.21 | –0.34 | –0.23 (–1.98, +1.67) |
| Management | +2.76 | –0.20 | –1.53 (–2.94, +0.17) |
| **Medicine** | | | |
| Genetics | +4.31 | +1.04 | +0.25 (–2.77, +3.00) |

*Appendix 1—table 2 Continued on next page*

*Appendix 1—table 2 Continued*

| Field | Observed $\Delta_{hiring}$ | Observed $\Delta_{attrition}$ | Est. $\Delta_{gendered\ attrition}$ |
|---|---|---|---|
| Pharmaceutical Sciences | +6.88 | –0.34 | –1.42 (–3.78, +0.96) |
| Epidemiology | +3.95 | +0.29 | –0.67 (–3.04, +1.59) |
| Pharmacology | +2.91 | +0.61 | –0.38 (–2.19, +1.36) |
| Pharmacy | +10.25 | –1.54 | –1.36 (–4.17, +1.52) |
| Physiology | +5.21 | +0.26 | –1.19 (–3.05, +0.72) |
| Veterinary Medical Sciences | +10.34 | –1.92 | –2.52 (–4.31, –0.74) |
| Immunology | +3.80 | +2.23 | +0.41 (–1.55, +2.35) |
| Natural Sciences | | | |
| Entomology | +6.49 | +0.85 | –1.42 (–4.55, +1.21) |
| Soil Science | +4.64 | +2.10 | +0.52 (–2.46, +3.38) |
| Anatomy | +6.05 | –0.82 | –1.61 (–4.11, +0.84) |
| Natural Resources | +4.70 | +1.64 | +0.09 (–2.23, +2.49) |
| Plant Sciences | +4.74 | +2.19 | +1.01 (–1.61, +3.68) |
| Plant Pathology | +4.88 | +1.51 | –1.70 (–4.92, +1.69) |
| Biophysics | +4.00 | –0.41 | –0.75 (–3.31, +1.71) |
| Food Science | +4.16 | +0.51 | –1.08 (–4.07, +2.03) |
| Pathology | +5.25 | –2.58 | –1.50 (–3.14, +0.04) |
| Horticulture | +2.60 | –0.12 | –1.66 (–5.07, +1.69) |
| Biostatistics | +3.58 | –0.69 | –0.68 (–3.49, +2.02) |
| Agronomy | +4.82 | +1.18 | –0.93 (–3.96, +1.68) |
| Animal Sciences | +7.46 | +1.74 | –0.51 (–2.68, +1.63) |
| Forestry and Forest Resources | +6.54 | +0.37 | –1.82 (–4.83, +0.71) |
| Geology | +5.93 | +1.54 | –0.38 (–1.85, +1.03) |
| Biological Sciences | +5.22 | +1.69 | –0.04 (–1.07, +0.92) |
| Physics | +2.32 | +1.00 | –0.25 (–1.00, +0.50) |
| Chemistry | +3.96 | +0.40 | –0.87 (–1.77, +0.02) |
| Biochemistry | +3.91 | +0.25 | –0.89 (–2.22, +0.40) |
| Chemical Engineering | +4.08 | +0.21 | –0.87 (–2.38, +0.63) |
| Environmental Sciences | +5.88 | +1.56 | –0.64 (–2.62, +1.16) |
| Atmospheric Sciences and Meteorology | +4.59 | +2.42 | +1.30 (–1.00, +3.30) |
| Biomedical Engineering | +4.29 | +2.08 | +0.44 (–1.38, +2.40) |
| Microbiology | +5.28 | +0.32 | –0.99 (–2.51, +0.59) |
| Cell Biology | +4.82 | +0.33 | –0.48 (2.06, +1.11) |
| Marine Sciences | +5.20 | +1.95 | –0.08 (–2.85, +2.44) |
| Astronomy | +2.89 | +1.52 | +0.08 (–1.22, +1.35) |
| Evolutionary Biology | +6.79 | +1.88 | +0.54 (–2.10, +3.21) |
| Ecology | +6.23 | +3.54 | +1.42 (–0.88, +3.92) |
| Neuroscience | +4.19 | –0.07 | –0.75 (–2.63, +1.02) |

*Appendix 1—table 2 Continued on next page*

*Appendix 1—table 2 Continued*

| Field | Observed $\Delta_{hiring}$ | Observed $\Delta_{attrition}$ | Est. $\Delta_{gendered\ attrition}$ |
|---|---|---|---|
| Molecular Biology | +3.40 | +0.83 | –0.16 (–1.83, +1.53) |
| **Mathematics and Computing** | | | |
| Statistics | +2.89 | +1.64 | +0.24 (–1.43, +1.72) |
| Mathematics | +2.86 | +0,65 | –0.44 (–1.22, +0.41) |
| Computer Engineering | +2.66 | +0.23 | –0.18 (–1.08, +0.79) |
| Computer Science | +2.26 | –0.09 | –0.55 (–1.51, +0.32) |
| Information Technology | +0.88 | –1.17 | –0.29 (–2.68, +2.01) |
| Information Science | +0.81 | –1.00 | –1.70 (–4.20, +1.05) |
| **Engineering** | | | |
| Mechanical Engineering | +3.39 | +0.60 | –0.18 (–1.09, +0.75) |
| Systems Engineering | +3.29 | +0.48 | –1.29 (–3.85, +1.15) |
| Aerospace Engineering | +2.53 | +1.18 | +0.33 (–1.27, +1.93) |
| Electrical Engineering | +2.18 | +0.46 | –0.22 (–1.11, +0.66) |
| Agricultural Engineering | +4.74 | +1.22 | –0.16 (–1.96, +1.44) |
| Operations Research | +2.53 | –0.39 | –0.99 (–3.44, +1.32) |
| Environmental Engineering | +4.94 | +0.91 | –0.91 (–2.30, +0.52) |
| Civil Engineering | +4.30 | +1.54 | –0.25 (–1.51, +1.07) |
| Materials Engineering | +4.68 | +0.46 | –0.85 (–2.76, +0.80) |
| Industrial Engineering | +1.86 | +1.66 | +0.14 (285, –2.25, +2.54) |

**Appendix 1—table 3.** Number of faculty by field and gender, 2020.
Estimated counts of women and men faculty based on 2020 faculty rosters and name-based gender inference (***Van Buskirk et al., 2023***).

| Field | Women | Men | Pct. women |
|---|---|---|---|
| **Health** | | | |
| Environmental Health Sciences | 285 | 430 | 39.9 |
| Nursing | 3531 | 515 | 87.3 |
| Public Health | 1813 | 1555 | 53.3 |
| Human Development and Family Sciences | 765 | 486 | 61.2 |
| Speech and Hearing Sciences | 352 | 184 | 65.7 |
| Exercise Science, Kinesiology, Rehab, Health | 1612 | 1555 | 50.9 |
| Nutrition Sciences | 722 | 604 | 54.4 |
| Communication Disorders and Sciences | 450 | 215 | 67.7 |
| Health, Physical Education, Recreation | 356 | 427 | 45.5 |
| Social Work | 1308 | 696 | 65.3 |
| **Education** | | | |
| Education | 1301 | 857 | 60.3 |
| Special Education | 452 | 277 | 62.0 |
| Education Administration | 986 | 840 | 54.0 |

*Appendix 1—table 3 Continued on next page*

*Appendix 1—table 3 Continued*

| Field | Women | Men | Pct. women |
|---|---|---|---|
| Counselor Education | 566 | 405 | 58.3 |
| Curriculum and Instruction | 1204 | 654 | 64.3 |
| **Journalism, Media and Communication** | | | |
| Communication | 1054 | 1208 | 46.6 |
| Mass Communications and Media Studies | 947 | 1093 | 46.4 |
| **Humanities** | | | |
| Theological Studies | 324 | 958 | 25.3 |
| Asian Languages | 189 | 257 | 42.4 |
| Slavic Languages and Literatures | 198 | 186 | 51.6 |
| Classics and Classical Languages | 513 | 596 | 46.3 |
| French Language and Literature | 291 | 253 | 53.5 |
| Germanic Languages and Literatures | 244 | 264 | 48.0 |
| Theatre Literature, History and Criticism | 574 | 615 | 48.3 |
| Art History and Criticism | 1006 | 912 | 52.5 |
| Asian Studies | 237 | 344 | 40.8 |
| History | 2071 | 3008 | 40.8 |
| Urban and Regional Planning | 335 | 506 | 39.8 |
| Linguistics | 404 | 467 | 46.4 |
| English Language and Literature | 2968 | 2018 | 50.4 |
| Near and Middle Eastern Languages and Cultures | 160 | 268 | 37.4 |
| Music | 1239 | 2747 | 31.1 |
| Philosophy | 788 | 1773 | 30.8 |
| Religious Studies | 446 | 900 | 33.1 |
| Comparative Literature | 354 | 364 | 49.3 |
| Spanish Language and Literature | 438 | 409 | 51.7 |
| Architecture | 606 | 1205 | 33.5 |
| **Public Administration and Policy** | | | |
| Public Policy | 687 | 1146 | 37.5 |
| Public Administration | 446 | 645 | 40.9 |
| **Social Sciences** | | | |
| Sociology | 1501 | 1483 | 50.3 |
| Gender Studies | 474 | 82 | 85.3 |
| Anthropology | 1305 | 1291 | 50.3 |
| Political Science | 1345 | 2702 | 33.2 |
| International Affairs | 426 | 851 | 33.4 |
| Geography | 482 | 933 | 34.1 |
| Psychology | 2826 | 3215 | 46.8 |
| Agricultural Economics | 171 | 532 | 24.3 |

*Appendix 1—table 3 Continued*

| Field | Women | Men | Pct. women |
| --- | --- | --- | --- |
| Educational Psychology | 555 | 463 | 54.5 |
| Economics | 804 | 3039 | 20.9 |
| Criminal Justice and Criminology | 466 | 588 | 44.2 |
| Business | | | |
| Accounting | 536 | 1186 | 31.1 |
| Marketing | 516 | 1069 | 32.6 |
| Management Information Systems | 231 | 867 | 21.0 |
| Finance | 377 | 1503 | 20.1 |
| Business Administration | 546 | 1473 | 27.0 |
| Management | 880 | 2136 | 29.2 |
| Medicine | | | |
| Genetics | 324 | 612 | 34.6 |
| Pharmaceutical Sciences | 444 | 912 | 32.7 |
| Epidemiology | 778 | 747 | 51.0 |
| Pharmacology | 512 | 1268 | 28.8 |
| Pharmacy | 567 | 639 | 47.0 |
| Physiology | 620 | 1446 | 30.0 |
| Veterinary Medical Sciences | 956 | 1281 | 42.7 |
| Immunology | 677 | 1303 | 34.2 |
| Natural Sciences | | | |
| Entomology | 191 | 494 | 27.9 |
| Soil Science | 163 | 506 | 24.4 |
| Anatomy | 387 | 763 | 33.7 |
| Natural Resources | 340 | 877 | 27.9 |
| Plant Sciences | 250 | 656 | 27.6 |
| Plant Pathology | 166 | 446 | 27.1 |
| Biophysics | 223 | 689 | 24.5 |
| Food Science | 353 | 474 | 42.7 |
| Pathology | 1199 | 1868 | 39.1 |
| Horticulture | 117 | 394 | 22.9 |
| Biostatistics | 457 | 676 | 40.3 |
| Agronomy | 149 | 526 | 22.1 |
| Animal Sciences | 337 | 749 | 31.0 |
| Forestry and Forest Resources | 229 | 655 | 25.9 |
| Geology | 770 | 2098 | 26.8 |
| Biological Sciences | 1895 | 3656 | 34.1 |
| Physics | 800 | 4364 | 15.5 |
| Chemistry | 1024 | 3691 | 21.8 |

*Appendix 1—table 3 Continued*

| Field | Women | Men | Pct. women |
|---|---|---|---|
| Biochemistry | 1001 | 2884 | 25.8 |
| Chemical Engineering | 405 | 1692 | 19.3 |
| Environmental Sciences | 624 | 1416 | 30.6 |
| Atmospheric Sciences and Meteorology | 267 | 757 | 26.1 |
| Biomedical Engineering | 426 | 1284 | 24.9 |
| Microbiology | 884 | 1815 | 32.8 |
| Cell Biology | 844 | 1699 | 33.2 |
| Marine Sciences | 279 | 722 | 27.9 |
| Astronomy | 416 | 1937 | 17.7 |
| Evolutionary Biology | 293 | 521 | 36.0 |
| Ecology | 370 | 662 | 35.9 |
| Neuroscience | 721 | 1443 | 33.3 |
| Molecular Biology | 732 | 1669 | 30.5 |
| **Mathematics and Computing** | | | |
| Statistics | 474 | 1598 | 22.9 |
| Mathematics | 1072 | 4688 | 18.6 |
| Computer Engineering | 584 | 3581 | 14.0 |
| Computer Science | 885 | 4291 | 17.1 |
| Information Technology | 211 | 759 | 21.8 |
| Information Science | 404 | 723 | 35.8 |
| **Engineering** | | | |
| Mechanical Engineering | 562 | 3428 | 14.1 |
| Systems Engineering | 152 | 654 | 18.9 |
| Aerospace Engineering | 209 | 1364 | 13.3 |
| Electrical Engineering | 613 | 3914 | 13.5 |
| Agricultural Engineering | 378 | 1386 | 21.4 |
| Operations Research | 149 | 632 | 19.1 |
| Environmental Engineering | 517 | 1854 | 21.8 |
| Civil Engineering | 585 | 2217 | 20.9 |
| Materials Engineering | 340 | 1439 | 19.1 |
| Industrial Engineering | 212 | 838 | 20.2 |

