## [Editor Report · eLife assessment]

Efforts to increase the representation of women in academia have focussed on efforts to recruit more women and to reduce the attrition of women. This study - which is based on analyses of data on more than 250,000 tenured and tenure-track faculty from the period 2011-2020, and the predictions of counterfactual models - shows that hiring more women has a bigger impact than reducing attrition. The study is an **important** contribution to work on gender representation in academia, and the evidence in support of the findings is **convincing**.

---

## [Referee Report · Reviewer #1 (Public Review)]

Summary

This is an interesting paper that concludes that hiring more women will do more to improve the gender balance of (US) academia than improving the attrition rates of women (which are usually higher than men's). Other groups have reported similar findings, i.e. that improving hiring rates does more for women's representation than reducing attrition, but this study uses a larger than usual dataset that spans many fields and institutions so it is a good contribution to the field.

The paper is much improved and far more convincing as a result of the revisions made by the authors.

Strengths

A large data set with many individuals, many institutions and fields of research.

A good sensitivity analysis to test for potential model weaknesses.

Weaknesses

Only a single country with a very specific culture and academic system.

Complex model fitting with many steps and possible places for model bias.

---

## [Referee Report · Reviewer #3 (Public Review)]

Summary

This study investigates the roles of faculty hiring and attrition in influencing gender representation in U.S. academia. It uses a comprehensive dataset covering tenured and tenure-track faculty across various fields from 2011 to 2020. The study employs a counterfactual model to assess the impact of hypothetical gender-neutral attrition and projects future gender representation under different policy scenarios. The analysis reveals that hiring has a more significant impact on women's representation than attrition in most fields and highlights the need for sustained changes in hiring practices to achieve gender parity.

The revisions made by the authors have improved the paper.

Strengths

Overall, the manuscript offers significant contributions to understanding gender diversity in academia through its rigorous data analysis and innovative methodology.

The methodology is robust, employing extensive data covering a wide range of academic fields and institutions.

Weaknesses

The primary weakness of the study lies in its focus on U.S. academia, which may limit the generalizability of its findings to other cultural and academic contexts. Additionally, the counterfactual model's reliance on specific assumptions about gender-neutral attrition could affect the accuracy of its projections.

Additionally, the study assumes that whoever disappeared from the dataset is attrition in academia. While in reality, those attritions could be researchers who moved to another country or another institution that is not indexed by AA.

---

## [Author Response]

The following is the authors’ response to the original reviews.

Your editorial guidance, reviews, and suggestions have led us to make substantial changes to our manuscript. While we detail point-by-point responses in typical fashion below, I wanted to outline, at a high level, what we’ve done.

(1) Methods. Your suggestions led us to rethink our presentation of our methods, which are now described more cohesively in a new methods section in the main text.

(2) Model Validation & Robustness. Reviewers suggested various validations and checks to ensure that our findings were not, for instance, the consequence of a particular choice of parameter. These can be found in the supplementary materials.

(3) Data Cleaning & Inclusion/Exclusion. Finally, based on feedback, our new methods section fully describes the process by which we cleaned our original data, and on what grounds we included/excluded individual faculty records from analysis.

**eLife assessment**
Efforts to increase the representation of women in academia have focussed on efforts to recruit more women and to reduce the attrition of women. This study - which is based on analyses of data on more than 250,000 tenured and tenure-track faculty from the period 2011-2020, and the predictions of counterfactual models - shows that hiring more women has a bigger impact than reducing attrition. The study is an important contribution to work on gender representation in academia, and while the evidence in support of the findings is solid, the description of the methods used is in need of improvement.
**Reviewer #1 (Public Review):**
Summary and strengthsThis is an interesting paper that concludes that hiring more women will do more to improve the gender balance of (US) academia than improving the attrition rates of women (which are usually higher than men's). Other groups have reported similar findings but this study uses a larger than usual dataset that spans many fields and institutions, so it is a good contribution to the field.

We thank the reviewer for their positive assessment of the contributions of our work.

WeaknessesThe paper uses a mixture of mathematical models (basically Leslie matrices, though that term isn't mentioned here) parameterised using statistical models fitted to data. However, the description of the methods needs to be improved significantly. The author should consider citing Matrix Population Models by Caswell (Second Edition; 2006; OUP) as a general introduction to these methods, and consider citing some or all of the following as examples of similar studies performed with these models:Shaw and Stanton. 2012. Proc Roy Soc B 279:3736-3741Brower and James. 2020. PLOS One 15:e0226392James and Brower. 2022. Royal Society Open Science 9:220785 Lawrence and Chen. 2015.[http://128.97.186.17/index.php/pwp/article/view/PWP-CCPR-2015-008]Danell and Hjerm. 2013. Scientometrics 94:999-1006

We have expanded the description of methods in a new methods section of the paper which we hope will address the reviewer’s concerns.

We agree that our model of faculty hiring and attrition resembles Leslie matrices. In results section B, we now mention Leslie matrices and cite Matrix Population Models by Caswell, noting a few key differences between Leslie matrices and the model of hiring and attrition presented in this work. Most notably, in the hiring and attrition model presented, the number of new hires is not based on per-capita fertility constants. Instead, population sizes are predetermined fixed values for each year, precluding exponential population growth or decay towards 0 that is commonly observed in the asymptotic behavior of linear Leslie Matrix models.

We have additionally revised the main text to cite the listed examples of similar studies (we had already cited James and Brower, 2022). We thank the reviewer for bringing these relevant works to our attention.

The analysis also runs the risk of conflating the fraction of women in a field with gender diversity! In female-dominated fields (e.g. Nursing, Education) increasing the proportion of women in the field will lead to reduced gender diversity. This does not seem to be accounted for in the analysis. It would also be helpful to state the number of men and women in each of the 111 fields in the study.

We have carefully examined the manuscript and revised the text to correctly differentiate between gender diversity and women’s representation.

We have additionally added a table to the supplemental materials (Tab. S3) that reports the estimated number of men and women in each of the 111 fields.

**Reviewer #2 (Public Review):**
Summary:This important study by LaBerge and co-authors seeks to understand the causal drivers of faculty gender demographics by quantifying the relative importance of faculty hiring and attrition across fields. They leverage historical data to describe past trends and develop models that project future scenarios that test the efficacy of targeted interventions. Overall, I found this study to be a compelling and important analysis of gendered hiring and attrition in US institutions, and one that has wide-reaching policy implications for the academy. The authors have also suggested a number of fruitful future avenues for research that will allow for additional clarity in understanding the gendered, racial, and socioeconomic disparities present in US hiring and attrition, and potential strategies for mitigating or eliminating these disparities.

We thank the reviewer for their positive assessment of the contributions of our work.

Strengths:In this study, LaBerge et al use data from over 268,000 tenured and tenure-track faculty from over 100 fields at more than 12,000 PhD-granting institutions in the US. The period they examine covers 2011-2020. Their analysis provides a large-scale overview of demographics across fields, a unique strength that allows the authors to find statistically significant effects for gendered attrition and hiring across broad areas (STEM, non-STEM, and topical domains).LaBerge et al. find gendered disparities in attrition-using both empirical data and their counterfactual model-that account for the loss of 1378 women faculty across all fields between 2011 and 2020. It is true that "this number is both a small portion of academia... and a staggering number of individual careers," as ." - as this loss of women faculty is comparable to losing more than 70 entire departments. I appreciate the authors' discussion about these losses-they note that each of these is likely unnecessary, as women often report feeling that they were pushed out of academic jobs.LaBerge et al. also find-by developing a number of model scenarios testing the impacts of hiring, attrition, or both-that hiring has a greater impact on women's representation in the majority of academic fields in spite of higher attrition rates for women faculty relative to men at every career stage. Unlike many other studies of historical trends in gender diversity, which have often been limited to institution-specific analyses, they provide an analysis that spans over 100 fields and includes nearly all US PhD-granting institutions. They are able to project the impacts of strategies focusing on hiring or retention using models that project the impact of altering attrition risk or hiring success for women. With this approach, they show that even relatively modest annual changes in hiring accumulate over time to help improve the diversity of a given field. They also demonstrate that, across the model scenarios they employ, changes to hiring drive the largest improvement in the long-term gender diversity of a field.Future work will hopefully - as the authors point out - include intersectional analyses to determine whether a disproportionate share of lost gender diversity is due to the loss of women of color from the professoriate. I appreciate the author's discussion of the racial demographics of women in the professoriate, and their note that "the majority of women faculty in the US are white" and thus that the patterns observed in this study are predominately driven by this demographic. I also highly appreciate their final note that "equal representation is not equivalent to equal or fair treatment," and that diversifying hiring without mitigating the underlying cause of inequity will continue to contribute to higher losses of women faculty.WeaknessesFirst, and perhaps most importantly, it would be beneficial to include a distinct methods section. While the authors have woven the methods into the results section, I found that I needed to dig to find the answers to my questions about methods. I would also have appreciated additional information within the main text on the source of the data, specifics about its collection, inclusion and exclusion criteria for the present study, and other information on how the final dataset was produced. This - and additional information as the authors and editor see fit - would be helpful to readers hoping to understand some of the nuance behind the collection, curation, and analysis of this important dataset.

We have expanded upon the description of methods in a new methods section of the paper.

We have also added a detailed description of the data cleaning steps taken to produce the dataset used in these analyses, including the inclusion/exclusion criteria applied. This detailed description is at the beginning of the methods section. This addition has substantially enhanced the transparency of our data cleaning methods, so we thank the reviewer for this suggestion.

I would also encourage the authors to include a note about binary gender classifications in the discussion section. In particular, I encourage them to include an explicit acknowledgement that the trends assessed in the present study are focused solely on two binary genders - and do not include an analysis of nonbinary, genderqueer, or other "third gender" individuals. While this is likely because of the limitations of the dataset utilized, the focus of this study on binary genders means that it does not reflect the true diversity of gender identities represented within the professoriate.In a similar vein, additional context on how gender was assigned on the basis of names should be added to the methods section.

We use a free, open-source, and open-data python package called nomquamgender (Van Buskirk et al, 2023) to estimate the strengths of (culturally constructed) name-gender associations. For sufficiently strong associations with a binary gender, we apply those labels to the names in our data. We have updated the main text to make this approach more apparent.

We have also added language to the main text which explicitly acknowledges that our approach only assigns binary (woman/man) labels to faculty. We point out that this is a compromise due to the technical limitations of name-based gender methodologies and is not intended to reinforce a gender binary.

I do think that some care might be warranted regarding the statement that "eliminating gendered attrition leads to only modest changes in field-level diversity" (Page 6). while I do not think that this is untrue, I do think that the model scenarios where hiring is "radical" and attrition is unchanged from present (equal representation of women and men among hires (ER) + observed attrition (OA)) shows that a sole focus on hiring dampens the gains that can otherwise be addressed via even modest interventions (see, e.g., gender-neutral attrition (GNA) + increasing representation of women among hires (IR)). I am curious as to why the authors did not include an additional scenario where hiring rates are equal and attrition is equalized (i.e., GNA + ER). The importance of including this additional model is highlighted in the discussion, where, on Page 7, the authors write: "In our forecasting analysis, we find that eliminating the gendered attrition gap, in isolation, would not substantially increase representation of women faculty in academia. Rather, progress towards gender parity depends far more heavily on increasing women's representation among new faculty hires, with the greatest change occurring if hiring is close to gender parity." I believe that this statement would be greatly strengthened if the authors can also include a comparison to a scenario where both hiring and attrition are addressed with "radical" interventions.

Our rationale for omitting the GNA + ER scenario in the presented analysis is that we can reason about the outcomes of this scenario without the need for computation; if a field has equal inputs of women and men faculty (on average) and equal retention rates between women and men (on average), then, no matter the field’s initial age and gender distribution of faculty, the expected value for the percentage of women faculty after all of the prior faculty have retired (which may take 40+ years) is exactly 50%. We have updated the main text to discuss this point.

**Reviewer #3 (Public Review):**
This manuscript investigates the roles of faculty hiring and attrition in influencing gender representation in US academia. It uses a comprehensive dataset covering tenured and tenure-track faculty across various fields from 2011 to 2020. The study employs a counterfactual model to assess the impact of hypothetical gender-neutral attrition and projects future gender representation under different policy scenarios. The analysis reveals that hiring has a more significant impact on women's representation than attrition in most fields and highlights the need for sustained changes in hiring practices to achieve gender parity.Strengths:Overall, the manuscript offers significant contributions to understanding gender diversity in academia through its rigorous data analysis and innovative methodology.The methodology is robust, employing extensive data covering a wide range of academic fields and institutions.Weaknesses:The primary weakness of the study lies in its focus on US academia, which may limit the generalizability of its findings to other cultural and academic contexts.

We agree that the U.S. focus of this study limits the generalizability of our findings. The findings that we present in this work will only generalize to other populations–whether it be to an alternate industry, e.g., tech workers, or to faculty in different countries–to the extent that these other populations share similar hiring patterns, retention patterns, and current demographic representation. We have added a discussion of this limitation to the manuscript.

Additionally, the counterfactual model's reliance on specific assumptions about gender-neutral attrition could affect the accuracy of its projections.

Our projection analysis is intended to illustrate the potential gender representation outcomes of several possible counterfactual scenarios, with each projection being conditioned on transparent and simple assumptions. In this way, the projection analysis is not intended to predict or forecast the future.

To resolve this point for our readers, we now introduce our projections in the context of the related terms of prediction and forecast, noting that they have distinct meanings as terms of art: On one hand, prediction and forecasting involve anticipating a specific outcome based on available information and analysis, and typically rely on patterns, trends, or historical data to make educated guesses about what will happen. Projections are based on assumptions and are often presented in a panel of possible future scenarios. While predictions and forecasts aim for precision, projections (which we make in our analysis) are more generalized and may involve a range of potential outcomes.

Additionally, the study assumes that whoever disappeared from the dataset is attrition in academia. While in reality, those attritions could be researchers who moved to another country or another institution that is not included in the AARC (Academic Analytics Research Centre) dataset.

In our revision, we have elevated this important point, and clarified it in the context of the various ways in which we count hires and attritions. We now explicitly state that “We define faculty hiring and faculty attrition to include all cases in which faculty join or leave a field or domain within our dataset.” Then, we enumerate the number of situations that could be counted as hires and attritions, including the reviewer’s example of faculty who move to another country.

**Reviewer #1 (Recommendations For The Authors):**
Section B: The authors use an age structured Leslie matrix model (see Caswell for a good reference to these) to test the effect of making the attrition rates or hiring rates equal for men and women. My main concern here is the fitting techniques for the parameters. These are described (a little too!) briefly in section S1B. Some specific questions that are left hanging include:A 5th order polynomial is an interesting choice. Some statistical evidence as to why it was the best fit would be useful. What other candidate models were compared? What was the "best fit" judgement made with: AIC, r^2? What are the estimates for how good this fit is? How many data points were fitted to? Was it the best fit choice for all of the 111 fields for men and women?

We use a logistic regression model for each field to infer faculty attrition probabilities across career ages and time, and we include the career age predictor up to its fifth power to capture the career-age correlations observed in Spoon et. al., Science Advances, 2023. For ease of reference, we reproduce the attrition risk curves in Fig S4.

We note that faculty attrition rates start low and then reach a peak around 5-7 years after earning PhD, and then decline until around 15-20 years post-PhD, after which, attrition rates increase as faculty approach retirement.

This function shape starts low and ends high, and includes at least one local minimum, which indicates that career age should be odd-ordered in the model and at least order-3, but only including career age up to its 3rd order term tended to miss some of the overserved career-age/attrition correlations. We evaluated the fit using 5-fold cross validation with a Brier score loss metric, and among options of polynomials of degree 1, 3, 5, or 7, we found that 5th order performed well overall on average over all fields (even if it was not the best for every field), without overfitting in fields with fewer data. Example fits, reminiscent of the figure from Spoon et al, are now provided in Figs S4 and S5.

While the model fit with fifth order terms may not be the best fit for all 111 fields (e.g., 7th order fits better in some cases), we wanted to avoid field-specific curves that might be overfitted to the field-specific data, especially due to low sample size (and thus larger fluctuations) on the high career age side of the function. Our main text and supplement now includes justifications for our choice to include career age up to its fifth order terms.

You used the 5th order logistic regression (bottom of page 11) to model attrition at different ages. The data in [24] shows that attrition increases sharply, then drops then increases again with career age. A fifth order polynomial on its own could plausibly do this but I associate logistic regression models like this as being monotonically increasing (or decreasing!), again more details as to how this worked would be useful.

Our first submission did not explain this point well, but we hope that Supplementary Figures S4 and S5 provide clarity. In short, we agree of course that typical logistic regression assumes a linear relationship between the predictor variables and the log odds of the outcome variable. This means that the relationship between the predictor variables and the probability of the outcome variable follows a sigmoidal (S-shaped) curve. However, the relationship between the predictor variables and the outcome variable may not be linear.

To capture more complex relationships, like the increasing, decreasing and then increasing attrition rates as a function of career age, higher-order terms can be added to the logistic regression model. These higher-order terms allow the model to capture nonlinear relationships between the predictor variables and the outcome variable — namely the non-monotonic relationship between rates of attrition and career age — while staying within a logistic regression framework.

"The career age of new hires follows the average career age distribution of hires" did you use the empirical distribution here or did you fit a standard statistical distribution e.g. Gamma?

We used the empirical distribution. This information has been added to the updated methods section in the main text.

How did you account for institution (presumably available)? Your own work has shown that institution types plays a role which could be contributing to these results.

See below.

What other confounding variables could be at play here, what is available as part of the data and what happens if you do/don't account for them?

A number of variables included in our data have been shown to correlate with faculty attrition, including PhD prestige, current institution prestige, PhD country, and whether or not an individual is a “self-hire,” i.e., trained and hired at the same institution (Wapman et. al., Nature, 2022). Additional factors that faculty self-report as reasons for leaving academia include issues of work-life balance, workplace climate, and professional reasons, and in some cases to varying degrees between men and women faculty (Spoon et. al., Sci. Adv., 2023).

Our counterfactual analysis aims to address a specific question: how would women’s representation among faculty be different today if men and women were subjected to the same attrition patterns over the past decade? To answer this question, it is important to account for faculty career age, which we accept as a variable that will always correlate strongly with faculty attrition rates, as long as the tenure filter remains in place and faculty continue to naturally progress towards retirement age. On the other hand, it is less clear why PhD country, self-hire status, or any of the other mentioned variables should necessarily correlate with attrition rates and with gendered differences in attrition rates more specifically. While some or all of these variables may underlie the causal roots of gendered attrition rates, our analysis does not seek to answer causal questions about why faculty leave their jobs (e.g., by testing the impact of accounting for these variables in simulations per the reviewers suggestion). This is because we do not believe the data used in this analysis is sufficient to answer such questions, lacking comprehensive data on faculty stress (Spoon et. al., Sci. Adv., 2023), parenthood status, etc.

What career age range did the model use?

The career age range observed in model outcomes are a function of the empirically derived attrition rates for faculty across academic fields. The highest career age observed in the AARC data was 80, and the faculty career ages that result from our model simulations and projections do not exceed 80.

We have also added the distribution of faculty across career ages for the projection scenario model outputs in the supplemental materials Fig. S3 (see response to your later comment regarding career age for further details). Looking at these distributions, it is observed that very few faculty have career age > 60, both in observation and in our simulations.

What was the initial condition for the model?

Empirical 2011 Faculty rosters are used as the initial conditions for the counterfactual analysis, and 2020 faculty rosters are these as the initial conditions for the projections analysis. This information has been added to the descriptions of methods in the main text.

Starting the model in 2011 how well does it fit the available data up to 2020?

Thank you for this suggestion. We ran this analysis for each field starting in 2011, and found that model outcomes were statistically indistinguishable from the observed 2020 faculty gender compositions for all 111 academic fields. This finding is not surprising, because the model is fit to the observed data, but it serves to validate the methods that we used to extract the model's parameters. We have added these results to the supplement (Fig. S2).

What are the sensitivity analysis results for the model? If you have made different fitting decisions how much would the results change? All this applied to both the hiring and attrition parameters estimates.

We model attrition and hiring using logistic regression, with career age included as an exogenous variable up to its fifth power. A natural question follows: what if we used a model with career age only to its first or third power? Or to higher powers? We performed this sensitivity analysis, and added three new figures to the supplement to present these findings:

First, we show the observed attrition probabilities at each career age, and four model fits to attrition data (Supplementary Figs S4 and S5). The first model includes career age only to its first power, and this model clearly does not capture the full career age / attrition correlation structure. The second model includes career age to its third power, which does a better job of fitting to the observed patterns. The third model includes career age up to its fifth power, which appears to very modestly improve upon the former model. The fourth model includes career age up to its seventh power, and the patterns captured by this model are largely the same as the 5th-power model up to career age 50, beyond which there are some notable differences in the inferred attrition probabilities. These differences would have relatively little impact on model outcomes because the vast majority of faculty have a career age below 50.

Second, we show the observed probability that hires are women, conditional on the career age of the hire. Once again, we fit four models to the data, and find that career age should be included at least up to its fifth order in order to capture the correlation structures between career age and the gender of new hires. However, limited differences result from including career age up to the 7th degree in the model (relative to the 5th degree).

As a final sensitivity analysis, we reproduce Fig. 2, but rather than including career age as an exogenous variable up to its fifth power in our models for hiring and attrition, we include career age up to its third power. Findings under this parameterization are qualitatively very similar to those presented in Fig. 2, indicating that the results are robust to modest changes to model parameterization (shown in supplement Fig. S6).

Far more detail in this and some interim results from each stage of the analysis would make the paper far more convincing. It currently has an air of "black box" too much of the analysis which would easily allow an unconvinced reader to discard the results.

We have added more detailed descriptions of the methods to the main text. We hope that the changes made will address these concerns.

Section C: You use the Leslie model to predict the future population. As the model is linear the population will either grow exponentially (most likely) or dwindle to zero. You mention you dealt with this by scaling the average value of H to keep the population at 2020 levels? This would change the ratio of hiring to attrition. How did this affect the timescale of the results. If a field had very minimal attrition (and hence grew massively over the time period of the dataset) the hiring rate would have to be very small too so there would be very little change in the gender balance. Did you consider running the model to steady state instead?

We chose the 40 year window (2020-2060) for this projection analysis because 40 years is roughly the timespan of a full-length faculty career. In other words, it will take around 40 years for most of the pre-existing faculty from 2020 to retire, such that the new, simulated faculty will have almost entirely replaced all former faculty by 2060.

For three out of five of our projection scenarios (OA, GNA, OA+ER), the point at which observed faculty are replaced by simulated faculty represents steady state. One way to check this intuition is to observe the asymptotic behavior of the trajectories in Fig. 3B; the slopes for these 3 scenarios nearly level out within 40 years.

The other two scenarios (OA + IR, GNA+IR) represent situations where women’s representation among new hires is increasing each year. These scenarios will not reach steady state until women represent 100% of faculty. Accordingly, the steady state outcomes for these scenarios would yield uninteresting results; instead, we argue that it is the relative timescales that are interesting.

What did you do to check that your predictions at least felt realistic under the fitted parameters? (see above for presenting the goodness of fit over the 10 years of the data).

We ran the analysis suggested in a prior comment (Starting the model in 2011 how well does it fit the available data up to 2020?) and found that model outcomes were statistically indistinguishable from the observed 2020 faculty gender compositions for all 111 academic fields, plus the “All STEM” and “All non-STEM” aggregations.

You only present the final proportion of women for each scenario. As mentioned earlier, models of this type have a tendency to lead to strange population distributions with wild age predictions and huge (or zero populations). Presenting more results here would assuage any worries the reader had about these problems. What is the predicted age distribution of men and women in the long term scenarios? Would a different method of keeping the total population in check have yielded different results? Interim results, especially from a model as complex as this one, rather than just presenting a final single number answer are a convincing validation that your model is a good one! Again, presenting this result will go a long way to convincing readers that your results are sound and rigorous.

Thank you for this suggestion. We now include a figure that presents faculty age distributions for each projection scenario at 2060 against the observed faculty age distribution in 2020 (pictured below, and as Fig. S3 in the supplementary materials). We find that the projected age distributions are very similar to the observed distributions for natural sciences (shown) and for the additional academic domains. We hope this additional validation will inspire confidence in our model of faculty hiring and attrition for the reviewer, and for future readers.

In Fig S3, line widths for the simulated scenarios span the central 95% of simulations.

Other people have reached almost identical conclusions (albeit it with smaller data sets) that hiring is more important than attrition. It would be good to compare your conclusions with their work in the Discussion.

We have revised the main text to cite the listed examples of similar studies. We thank the reviewer for bringing these relevant works to our attention.

General comments:What thoughts have you given to non-binary individuals?Be careful how you use the term "gender diversity"! In many countries "Gender diverse" is a term used in data collection for non-binary individuals, i.e. Male, female, gender diverse. The phrase "hiring more gender diverse faculty" can be read in different ways! If you are only considering men and women then gender balance may be a better framework to use.

We have added language to the main text which explicitly acknowledges that our analysis focuses on men and women due to limitations in our name-based gender tool, which only assigns binary (woman/man) labels to faculty. We point out that this is a compromise due to the technical limitations of name-based gender methodologies and is not intended to reinforce a gender binary.

We have also taken additional care with referring to “gender diversity,” per reviewer 1’s point in their public review.

**Reviewer #2 (Recommendations For The Authors):**
Data availability: I did not see an indication that the dataset used here is publicly available, either in its raw format or as a summary dataset. Perhaps this is due to the sensitive nature of the data, but regardless of the underlying reason, the authors should include a note on data availability in the paper.

The dataset used for these analyses were obtained under a data use agreement with the Academic Analytics Research Center (AARC). While these data are not publicly available, researchers may apply for data access here: https://aarcresearch.com/access-our-data.

We also added a table to the supplemental materials (Tab. S3) that reports the estimated number of men and women in each of the 111 fields.

Additionally, a variety of summary statistics based on this dataset are available online, here: https://github.com/LarremoreLab/us-faculty-hiring-networks/tree/main

Gender classification: Was an existing package used to classify gender from names in the dataset, or did the authors develop custom code to do so? Either way, this code should be cited. I would also be curious to know what the error rate of these classifications are, and suggest that additional information on potential biases that might result from automated classifications be included in the discussion, under the section describing data limitations. The reliability of name-based gender classification is particularly of interest, as external gender classifications such as those applied on the basis of an individual's name - may not reflect the gender with which an individual self-identifies. In other words, while for many people their names may reflect their true genders, for others those names may only reflect their gender assigned at birth and not their self-perceived or lived gender identity. Nonbinary faculty are in particular invisibilized here (and through any analysis that assigns binary gender on the basis of name). While these considerations do not detract from the main focus of the study - which was to utilize an existing dataset classified only on the basis of binary gender to assess trends for women faculty-these limitations should be addressed as they provide additional context for the interpretation of the results and suggest avenues for future research.

We use a free, open-source, and open-data python package called nomquamgender (Van Buskirk et al, 2023) to estimate the strengths of (culturally constructed) name-gender associations. For sufficiently strong associations with a binary gender, we apply those labels to the names in our data. We have updated the main text to make this approach more apparent.

We have also added language to the main text which explicitly acknowledges that our approach only assigns binary (woman/man) labels to faculty. We point out that this is a compromise due to the technical limitations of name-based gender methodologies and is not intended to reinforce a gender binary.

As we mentioned in response to the public review, we use a free and open source python package called nomquamgender to estimate the strengths of name-gender associations, and we apply gender labels to the names with sufficiently strong associations with a binary gender. This package is based on a paper by Van Buskirk et. al. 2023, “An open-source cultural consensus approach to name-based gender classification,” which documents error rates and potential biases.

We have also added language to the main text which explicitly acknowledges that our approach only assigns binary (woman/man) labels to faculty. We point out that this is a compromise due to the technical limitations of name-based gender methodologies and is not intended to reinforce a gender binary.

Page 1: The sentence beginning "A trend towards greater women's representation could be caused..." is missing a conjunction. It should likely read: "A trend towards greater women's representation could be caused entirely by attrition, e.g., if relatively more men than women leave a field, OR entirely by hiring..."

We have edited the paragraph to remove the sentence in question.

Pages 1-2: The sentence beginning "Although both types of strategy..." and ending with "may ultimately achieve gender parity" is a bit of a run-on; perhaps it would be best to split this into multiple sentences for ease of reading.

We have revised this run-on sentence.

Page 2: See comments in the public review about a methods section, the addition of which may help to improve clarity for the readers. Within the existing descriptions of what I consider to be methods (i.e., the first three paragraphs currently under "results"), some minor corrections could be added here. First, consider citing the source of the dataset in the line where it is first described (in the sentence "For these analyses, we exploit a census-level dataset of employment and education records for tenured and tenure-track faculty in 12,112 PhD-granting departments in the United States from 2011-2020.") It also may be helpful to include context here (or above, in the discussion about institutional analyses) about how "departments" can be interpreted. For example, how many institutions are represented across these departments? More information on how the authors eliminated the gendered aspect of patterns in their counterfactual model would be helpful as well; this is currently hinted at on page 4, but could instead be included in the methods section with a call-out to the relevant supplemental information section (S2B).

We have added a citation to Academic Analytics Research Center’s (AARC) list of available data elements to the data’s introduction sentence. We hope this will allow readers to familiarize themselves with the data used in our analysis.

Faculty department membership was determined by AARC based on online faculty rosters. 392 institutions are represented across the 12,112 departments present in our dataset. We have updated the main text to include this information.

Finally, we have added a methods section to the main text, which includes information on how the gendered aspect of attrition patterns were eliminated in the counterfactual model.

Page 2: Perhaps some indication of how many transitions from an out-of-sample institution might be helpful to readers hoping to understand "edge cases."

In our analysis, we consider all transitions from out-of-sample institutions to in-sample institutions as hires, and all transitions away from in-sample institutions–whether it be to an out of sample institution, or out of academia entirely–as attritions. We choose to restrict our analysis of hiring and attrition to PhD granting institutions in the U.S. in this way because our data do not support an analysis of other, out-of-sample institutions.

I also would have liked additional information on how many faculty switched institutions but remained "in-sample and in the same field" - and the gender breakdowns of these institutional changes, as this might be an interesting future direction for studies of gender parity. (For example, readers may be spurred to ask: if the majority of those who move institutions are women, what are the implications for tenure and promotion for these individuals?)

While these mid-career moves are not counted as attritions in the present analysis, a study of faculty who switch institutions but remain (in-sample) as faculty could shed light on issues of gendered faculty retention at the level of institutions. We share the reviewer’s interest in a more in depth study of mid-career moves and how these moves impact faculty careers, and we now discuss the potential value of such a study towards the end of the paper. In fact, this subject is the topic of a current investigation by the authors!

Page 3: I was confused by the statement that "of the three types of stable points, only the first point represents an equitable steady-state, in which men and women faculty have equal average career lengths and are hired in unchanging proportions." Here, for example, computer science appears to be close to the origin on Figure 1, suggesting that hiring has occurred in "unchanging proportions" over the study interval. However, upon analysis of Table S2, it appears that changes in hiring in Computer Science (+2.26 pp) are relatively large over the study interval compared to other fields. Perhaps I am reading too literally into the phrase that "men and women faculty are hired in unchanging proportions" - but I (and likely others) would benefit from additional clarity here.

We had created an arrow along with the computer science label in Fig. 1, but it was difficult to see, which is likely the source of this confusion. This was our fault, and we have moved the “Comp. Sci.” label and its corresponding arrow to be more visible in Figure 1.

Changes in women’s representation in Computer Science due to hiring over 2011 - 2020 was +2.26 pp as the reviewer points out, but, consulting Fig. 1 and the corresponding table in the supplement, we observe that this is a relatively small amount of change compared to most fields.

Page 3: If possible it may be helpful to cite a study (or multiple) that shows that "changes in women's representation across academic fields have been mostly positive." What does "positive" mean here, particularly when the changes the authors observe are modest? Perhaps by "positive" you mean "perceived as positive"?

We used the term positive in the mathematical sense, to mean greater than zero. We have reworded the sentence to read “women's representation across academic fields has been mostly increasing…” We hope this change clarifies our meaning to future readers.

Page 3: The sentence that ends with "even though men are more likely to be at or near retirement age than women faculty due to historical demographic trends" may benefit from a citation (of either Figure S3 or another source).

We now cite the corresponding figure in this sentence.

Page 4: The two sentences that begin with "The empirical probability that a person leaves their academic career" would benefit from an added citation.

We have added a citation to the sentences.

Figure 3: Which 10 academic domains are represented in Panel 3B? The colors in appear to correspond to the legend in Panel 3A, but no indication of which fields are represented is provided. If possible, please do so - it would be interesting and informative to be able to make these comparisons.

This was not clear in the initial version of Fig. 3B, so we now label each domain. For reference, the domains represented in 3B are (from top to bottom):

● Health

● Education

● Journalism, Media, Communication

● Humanities

● Social Sciences

● Public Administration and Policy

● Medicine

● Business

● Natural Sciences

● Mathematics and Computing

● Engineering

Page 6: Consider citing relevant figure(s) earlier up in paragraph 2 of the discussion. For example, the first sentence could refer to Figure 1 (rather than waiting until the bottom of the paragraph to cite it).

Thank you for this suggestion, we now cite Fig. 1 earlier in this discussion paragraph.

Page 10: A minor comment on the fraction of women faculty in any given year-the authors assume that the proportion of women in a field can be calculated from knowing the number of women in a field and the number of men. This is, again, true if assuming binary genders but not true if additional gender diversity is included. It is likely that the number of nonbinary faculty is quite low, and as such would not cause a large change in the overall proportions calculated here, but additional context within the first paragraph of S1 might be helpful for readers.

We have added additional context in the first paragraph of S1, explaining that an additional term could be added to the equation to account for nonbinary faculty representation if our data included nonbinary gender annotations. Thank you for making this point.

Page 10: Please include a range of values for the residual terms of the decomposition of hiring and attrition in the sentence that reads "In Figure S1 we show that the residual terms are small, and thus the decomposition is a good approximation of the total change in women's representation."

These residual terms range from -0.51pp to 1.14pp (median = 0.2pp). We have added this information to the sentence in question.

Page 12: It may be helpful to readers to include a description of the information contained in Table S2 in the supplemental text under section S3.

We refer to table S2 twice in the main text (once in the observational findings, and once for the counterfactual analysis), and the contents of table S2 are described thoroughly in the table caption.

**Reviewer #3 (Recommendations For The Authors):**
(1) There is a potential limitation in the generalizability of the findings, as the study focuses exclusively on US academia. Including international perspectives could have provided a more global understanding of the issues at hand.

The U.S. focus of this study limits the generalizability of our findings, as non-U.S. other faculty may exhibit differences in hiring patterns, retention patterns, and current demographic representations. We have added a discussion of this limitation to the manuscript. Unfortunately, our data do not support international analyses of hiring and attrition.

(2) I am not sure that everyone who disappeared from the AARC dataset could be count as "attrition" from academia. Indeed, some who disappeared might have completely left academia once they disappeared from the AARC dataset. Yet, there's also the possibility that some professors left for academic positions in countries outside of the US, or US institutions that are not included in the AARC dataset. These individuals didn't leave academia. Furthermore, it is also possible that these scholars who moved to an institution outside of US or not indexed by AARC are gender specific. Therefore, analyses that this study conducts should find a way to test whether the assumption that anyone who disappeared from AARC is indeed valid. If not, how will this potentially challenge the current conclusions?

The reviewer makes an important point: faculty who move to faculty positions in other countries and faculty who move to non-PhD granting institutions, or to institutions that are otherwise not included in the AARC data are all counted as attritions in our analysis. We intentionally define hiring and attrition broadly to include all cases in which faculty join or leave a field or domain within our dataset.

The types of transitions that faculty make out of the tenure track system at PhD granting institutions in the U.S. may correlate with faculty attributes, like gender. For example, women or men may be more likely to transition to tenure track positions at non-U.S. institutions. Nevertheless, these types of career transition represent an attrition for the system of study, and a hire for another system. Following this same logic, faculty who transition from one field to another field in our analysis are treated as an attrition from the first field and a hire into the new field.

By focusing on “all-cause” attrition in this way, we are able to make robust insights for the specific systems we consider (e.g.,, STEM and non-STEM faculty at U.S. PhD granting institutions), without being roadblocked by the task of annotating faculty departures and arbitrating which should constitute “valid” attritions.

(3) It would be very interesting to know how much of the attribution was due to tenure failure. Previous studies have suggested that women are less likely to be granted tenure, which makes me wonder about the role that tenure plays in the gendered patterns of attrition in academia.

We note that faculty attrition rates start low and then reach a peak around 5-7 years after earning PhD, and then decline until around 15-20 years post-PhD, after which, attrition rates increase as faculty approach retirement. The first local maximum appears to coincide roughly with the tenure clock timing, but we can only speculate that these attritions are tenure related. Our dataset is unfortunately not equipped to determine the causal mechanisms driving attrition.

We reproduce the attrition risk curve in the supplementary materials, Fig. S4:

(4) The dataset used doesn't fully capture the complexities of academic environments, particularly smaller or less research-intensive institutions (regional universities, historically black colleges and universities, and minority-serving institutions). This could be potentially added to the manuscript for discussions.

We have added this point to the description of this study’s limitations in the discussion.